# Giant intrinsic circular dichroism of prolinol-derived squaraine thin films

Matthias Schulz[1], Jennifer Zablocki[1], Oliya S. Abdullaeva[2], Stefanie Brück[1], Frank Balzer [3], Arne Lützen [1], Oriol Arteaga[4] & Manuela Schiek [2]

Molecular chirality and the inherently connected differential absorption of circular polarized light (CD) combined with semiconducting properties offers great potential for chiral opto-electronics. Here we discuss the temperature-controlled assembly of enantiopure prolinol functionalized squaraines with opposite handedness into intrinsically circular dichroic, molecular J-aggregates in spincasted thin films. By Mueller matrix spectroscopy we accurately probe an extraordinary high excitonic circular dichroism, which is not amplified by mesoscopic ordering effects. At maximum, CD values of 1000 mdeg/nm are reached and, after accounting for reflection losses related to the thin film nature, we obtain a film thickness independent dissymmetry factor $g = 0.75$. The large oscillator strength of the corresponding absorption within the deep-red spectral range translates into a negative real part of the dielectric function in the spectral vicinity of the exciton resonance. Thereby, we provide a new small molecular benchmark material for the development of organic thin film based chiroptics.

[1] Kekulé Insitute of Organic Chemistry and Biochemistry, Rheinische-Friedrich-Wilhelms-University of Bonn, Gerhard-Domagk-Str. 1, D-53121 Bonn, Germany. [2] Energy and Semiconductor Research Laboratory, Institute of Physics, Carl-von-Ossietzky-University of Oldenburg, Carl-von-Ossietzky-Str. 9-11, D-26129 Oldenburg, Germany. [3] Mads Clausen Institute, University of Southern Denmark, Alsion 2, DK-6400 Sønderborg, Denmark. [4] Department of Applied Physics and IN2UB, University of Barcelona, Barcelona 08028, Spain. Correspondence and requests for materials should be addressed to M.S. (email: manuela.schiek@uni-oldenburg.de)

Chirality is a fundamental symmetry property describing an object such as a molecule that is non-superimposable on its mirror image. Nature has evolved with single handedness meaning that essentially all biological processes are intrinsically chiral[1]. This is well recognized for instance in pharmacological applications, but however, the enantiopure, i.e. homochiral, synthesis of drugs remains a challenge for chemists. Other applications such as chiral optoelectronics (chiroptics) not focused on the underlying mechanism of molecular recognition processes or structural ordering in optoelectronic thin films[2,3] are still rare. Optical activity comprising circular birefringence (CB) and circular dichroism (CD), is an inherent optical property of chiral molecules or chiral supramolecular assemblies. CD manifests in the differential absorption of left- and right-handed circular polarized (CP) light[4–6]. This effect can be utilized for light harvesting or emitting devices with direct detectivity[7] or emissivity[8] of CP light, respectively. Prerequisite are the availability of the enantiopure molecular components through an efficient and scalable synthesis, and the subsequent transfer of molecular chirality via supramolecular interactions to a sizable macroscopic scale[9–12]. Currently, (plasmonic) metamaterials[13], which incorporate structures with characteristic lengths comparable to the wavelength of light, surpass natural chiral materials regarding their chiroptical effects. For organic matter thin films without mesoscopic structural ordering, the layer thickness normalized ellipticity typically is below 10 mdeg/nm due to a weak overall absorbance, and the corresponding absorbance normalized dissymmetry factor $g$ ranges between $10^{-4}$ to $10^{-2}$ [14–17]. In rare cases, extreme $g$-values exceeding $0.3$[18] or even up to 1[8,19] have been reported for polyfluorene thin films. These values include thickness and orientation dependent, pseudo-CD effects originating from cross-terms between linear dichroism and birefringence[20] as well as selective CP reflection due to long-range cholesteric ordering (Bragg reflection)[21–23]. This liquid crystallinity results in amplified $g$-values[24]. Pseudo-CD effects are concomitantly probed for (anisotropic) thin film samples due to the oversimplified utilization of (commercial) CD-spectro-polarimeters, which are well-designed for diluted solutions and any other isotropic samples, but not for the quantitative assessment of CD of typical thin film samples[25,26].

Here we show the versatility of a rotating compensator ellipsometer to perform Mueller matrix spectroscopy[27] to attain a true CD spectrum of spin casted thin films from enantiopure prolinol-functionalized squaraine compounds. Following a previously published, updated ex-chiral pool strategy[28,29] we have obtained the two enantiomers (S,S)-ProSQ-C16 and (R,R)-ProSQ-C16, where C16 denotes a linear alkyl chain with sixteen carbon atoms attached to the prolinol functional group, see Supplementary Note 1 and Supplementary Fig. 1 for details. We show that thermal annealing of such thin films leads to a giant and sharp excitonic CD within the deep-red spectral range located around 780 nm. The normalized CD well reaches a value of 1000 mdeg/nm and a dissymmetry factor of $g = 0.75$ for highly enantiopure films. To obtain this layer thickness independent dissymmetry factor we propose a simple approach to account for reflection losses of the measured absorbance related to the sample's thin film nature. The circular dichroic properties are evenly distributed over the complete thin film area and most importantly are free from contributions due to mesoscopic structural ordering effects. The large oscillator strength of the optically active absorbance resonance causes the real part of the dielectric function to become negative over a small wavelength range just below this exciton resonance. To the best of our knowledge, this combination of negative real permittivity and CD has not been documented previously for excitonic organic materials. Thereby, we provide a new small molecular benchmark material for the development of organic thin film based chiroptics.

## Results

**Early stage supramolecular aggregation in solution.** To begin with, we investigate the early stage supramolecular aggregation and optical activity of ProSQ-C16 colloidal aggregates in solution by so-called poor solvent titration experiments. This helps us to develop a, yet phenomenological, picture of the aggregation in the thin films. The compound is dissolved in a good solvent, here chloroform, and this stock solution is diluted with a mixture of chloroform and a poor solvent, here acetonitrile, with increasing volume fraction of acetonitrile[28]. Subsequently, UV-Vis spectra and CD-spectra of the colloidal solution are recorded. The UV-Vis spectra of (S,S)-ProSQ-C16, Fig. 1a and (R,R)-ProSQ-C16, Fig. 1b, are qualitatively equivalent. The sharp monomer absorption centered at 645 nm, accompanied by a vibronic progression at 595 nm, stems from the electronic $S_0 \rightarrow S_1$ transition along the long molecular axis of the squaraine backbone[30]. It decreases with increasing acetonitrile fraction in favor of an aggregate peak absorbing at higher energy around 550 nm. No peak at longer wavelengths appears contrary to the thin films we discuss hereafter. The change in color of the solution is visible to the eye, see Fig. 1e. Both ProSQ-C16 monomers are CD-silent, Fig. 1c, d, but the aggregate species show clear bisignate CD-couplets crossing zero where the absorbance spectrum peaks, which is evident for excitonic CD[6,31,32]. No vibronic progressions are noticeable for the aggregated species, which is indicative for a strong excitonic coupling[33]. The prolinol functional groups serve as aggregation directing chiral centers for homochiral, helical packing of the achiral chromophores, which here are the squaraine backbones. The excitonic CD signals result from coupling of twisted side-by-side arranged molecules within a helical stack (intra-helical stack coupling). Together with the spectral blue shift we can clearly identify the aggregates as H-aggregates[34]. Helical $\pi$-stacking (H-type aggregation) of carotenoid assemblies[33,35], conjugated polymers[24,36,37] or molecular compounds[10,38–40] induced by chiral side-chains is a commonly adopted route to excitonic supramolecular chirality[41]. Formation of CD-active J-type aggregates in solution have been rarely documented[29,42]. In case of our (S,S)-enantiomer, Fig. 1c, negative excitonic chirality is visible, i.e. the Cotton effect is negative at longer wavelength. This is indicative for the formation of a counter clockwise (left-handed) molecular helix, as sketched in Fig. 1a. As expected for a mirror image, Fig. 1d, the (R,R)-enantiomer shows positive excitonic chirality, i.e. the Cotton effect is positive at longer wavelength. This suggests the formation of a clockwise (right-handed) molecular helix, as sketched in Fig. 1b. Note that these sketches are for illustration only, and do not necessarily provide the correct number of molecules involved in the coupling, torsional angles or intermolecular spacings. Within the range of experimental error the CD-spectra of both colloidal enantiomer solutions are mirror-imaged but otherwise equivalent. Any minor enantiomeric impurities (see Supplementary Note 2 and Supplementary Fig. 2) will not be noticeable, since preferably molecules of the same handedness assemble (chiral discrimination), and the concentration threshold for aggregation of the undesired enantiomer or diastereomer is not reached.

**Morphology and unpolarized optical properties of thin films.** In the next step, we discuss temperature induced aggregation and unpolarized optical properties of thin films. We prepare these films by spincasting from solution and subsequent thermal annealing. The temperature treatment ranging from no annealing to 210 °C does not induce any crystallization or formation of distinct morphological features with micro- to mesoscopic domain size, which is rather untypical for small molecules and in

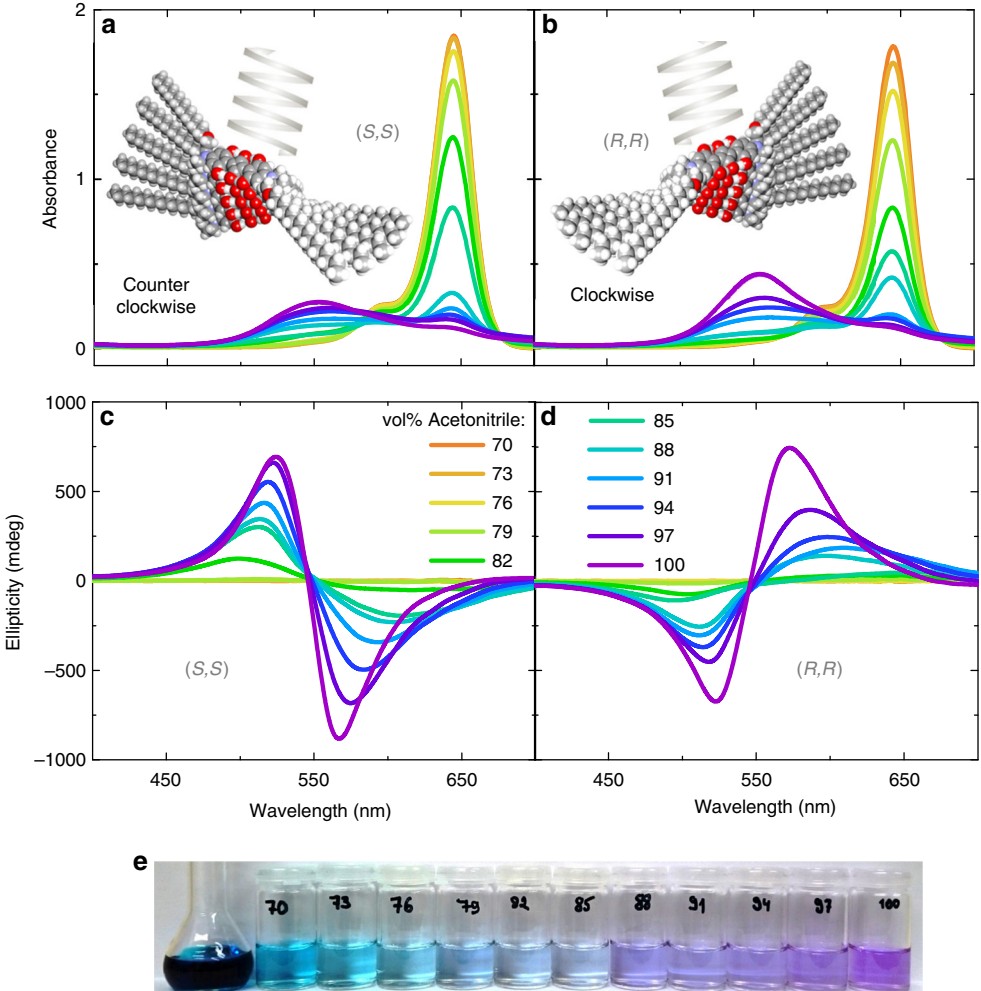

**Fig. 1** Early stage aggregation experiments in solution. **a**, **b** UV-vis spectra and **c,d** ellipticity spectra of (*S,S*)-ProSQ-C16 and (*R,R*)-ProSQ-C16 in mixtures of chloroform and acetonitrile, respectively. The concentration is approximately 5.7 μM. The volume fraction of acetonitrile is increased as indicated. The insets in **a** and **b** illustrate the proposed helical aggregation of the molecules as follows from the line shape of the CD bands. Note that these here are only sketches. In **e** a photograph of the stock solution and diluted solutions with increasing acetonitrile volume faction from left to right is shown

particular anilino squaraines[43]. For annealing at 240 °C a substantial dewetting of the organic material is seen leaving only few material droplets behind, see Supplementary Fig. 3. All other films are cohesive and free from long-range crystallographic order, which is concluded from X-ray diffraction Supplementary Fig. 4. In an optical microscope and in an atomic force microscope (AFM) the films look alike for all annealing temperatures up to 210 °C, Fig. 2. Clusters with heights of several tens of nanometers to a few hundreds of nanometers are distributed randomly over the faintly birefringent thin films. Small Maltese-cross-shaped birefringence patterns are noticeable around the clusters in crossed polarized optical microscopy images, Fig. 2d. Otherwise, no in-plane anisotropy is noticeable. Supplementary Movies 1 and 2 provide movies pieced together from series of overexposed bireflectance microscopy images of samples rotated between crossed polarizers for further illustration. The $R_{rms}$ surface roughness of the widespread film areas is about 6 nm. Thus the films are smooth enough to appear as dark background in darkfield microscopy imaging, Fig. 2c. Typically, three main distinct height levels can be identified on a sample with a spacing of 3 nm each, Fig. 2d. This step height could correspond to upright standing ProSQ-C16 molecules, or rather to the diameter of a helical coil of π-stacked molecules (cf. aggregation in solution), which is lying with its long coil axis parallel to the surface.

In Fig. 3a we show absolute absorbance spectra of ProSQ-C16 thin films without annealing and annealed at varying temperatures starting from 60 °C in steps of 30 °C up to 240 °C. The spectra for 120 °C, 180 °C and 210 °C annealing are identical within the range of experimental spread, therefore only the absorbance for 180 °C annealing is plotted in Fig. 3a for clarity of the presentation. The spectra of the (*R,R*)- and the (*S,S*)-enantiomer are indistinguishable. Above the graph photographs of selected samples are shown to give an impression of color. Unlike the morphology the absorbance properties clearly depend on temperature treatment. With increasing annealing temperature the long-wavelength peak is increasing in intensity and its maximum is red-shifting to 780 nm, while the broad short-wavelength feature is decreasing and narrowing with its maximum blue-shifting. Further increasing the annealing temperature results in disruption of the absorbance properties. According to the spectral positions relative to the monomer absorption maximum, Fig. 1, we assign the long-wavelength peak as J-band and the short-wavelength peak as H-band[44–47]. The Davydov splitting in between is large, estimated from the absorbance spectrum for 180 °C to be easily exceeding 0.4 eV.

Effective optical constants valid for both enantiomers of ProSQ-C16 thin films annealed at 180 °C have been obtained by fitting spectroscopic ellipsometry data. The real and the imaginary part of the complex refractive index, *n* and *k*, are

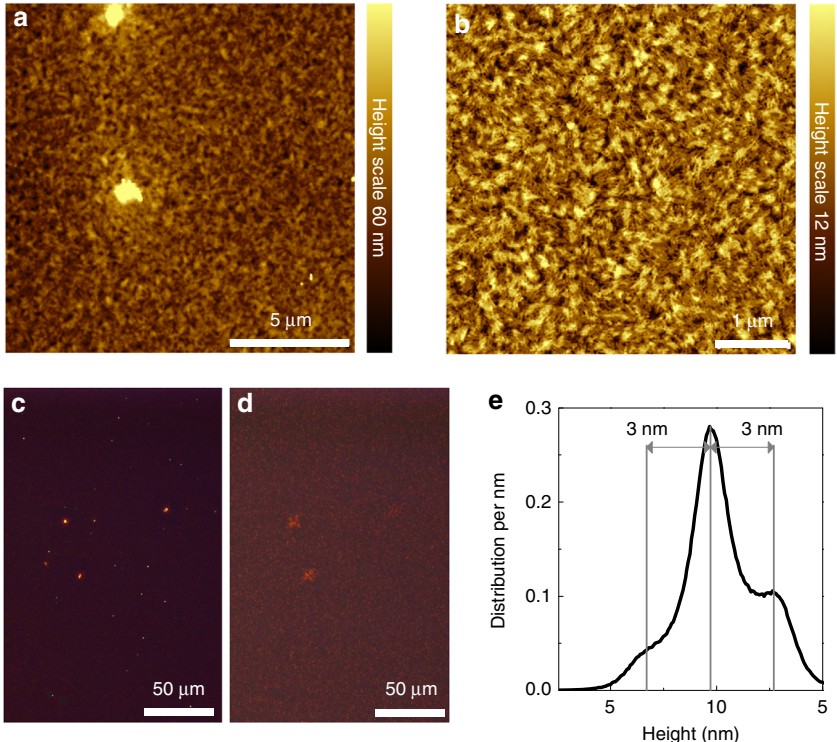

**Fig. 2** Micro-morphological characterization of ProSQ-C16 thin films. **a**, **b** Intermittent contact mode AFM images, and optical microscopy images **c** darkfield and **d** between crossed polarizers of typical ProSQ-C16 samples annealed at 180 °C. The optical microscopy images show the same sample area and are overexposed for better visibility. **e** Distribution of heights extracted from AFM image (**b**)

graphed in Supplementary Fig. 5. They reach remarkable maximum values of 4.5 and 3.3, respectively, at the J-band. The optical constants reproduce the transmission and reflection spectra of our thin films well. The real and the imaginary part of the dielectric function, $\varepsilon' = n^2 - k^2$ and $\varepsilon'' = 2nk$, are calculated based on the optical constants and plotted in Fig. 3b. In the spectral regime just below the excitonic J-band resonance at 780 nm the real part $\varepsilon'$ becomes negative due to the large oscillator strength of the resonance. This feature that made plasmonics prominent, albeit having a different physical origin, is underexplored for organic excitonic materials[48–51].

Obviously the ProSQ-C16 has a predominant tendency for J-type aggregation in thin films contrary to the colloidal aggregation in solution resulting in H-aggregates only, cf. Fig. 1. A similar behavior has been found for pseudocyanine, which is a textbook example for J-aggregates[52,53], that actually forms H-aggregates on the early steps of aggregation[54]. We propose that this spectral relocation in favor of the J-band is caused by inter-helical coupling between stacks in the solid state with increasing dominance upon increased thermal annealing[33]. On the molecular level, the interacting transition dipole moments of adjacent coils adopt an oblique-angled in-line orientation. The red-shift of the peak maximum can be caused by a flattening of the orientational angle for increased in-line coupling of the transition dipoles. The asymmetric broadening of the peak towards smaller wavelength might stem from a distribution of the obliqueness angle. In general, it is typical for squaraine thin films on a glass support that the intermolecular interactions dominate over the molecule-substrate interactions, so that the molecular $\pi$-stacking direction is orientated parallel to the substrate[43]. In case of ProSQ-C16 this means that the long helical molecular stacking axis is oriented within the plane of the substrate, but with random in-plane alignment consistent with the lack of linear dichroism. Note that this is a phenomenological model that requires

validation by a structural model based on quantum chemical calculations. However, the circular dichroic properties of the thin films discussed in the following support this preliminary model.

**Circular dichroic optical properties of thin films.** We utilized a VASE with a single compensator in a polarizer-compensator-sample-analyser (PCSA) configuration to perform Mueller matrix spectroscopy. Such an instrument can record 11, i.e. the first three rows out of the 15 elements of a normalized Mueller matrix. We can deduct the missing fourth row of the matrix by symmetry considerations to obtain a complete (4 × 4) Mueller matrix **M**, as it was verified for selected samples in a full Mueller matrix ellipsometer. Details on the experiment are given in the Methods section. After calculating the matrix logarithm of **M**, the differential Mueller matrix **L** has the following symmetry for non-depolarizing samples:[55–57]

$$\mathbf{L} = \ln(\mathbf{M}) = \begin{pmatrix} 0 & -LD & -LD' & CD \\ -LD & 0 & CB & LB' \\ -LD' & -CB & 0 & -LB \\ CD & -LB' & LB & 0 \end{pmatrix} \quad (1)$$

For transmission of light along a homogeneous medium the six independent parameters have the following physical interpretation: CD, circular dichroism; CB, circular birefringence; LD, horizontal linear dichroism; LB, horizontal linear birefringence; LD', 45° linear dichroism; LB', 45° linear birefringence.

The circular dichroism value CD is extracted from the differential Mueller matrix elements by:

$$CD = (l_{03} + l_{30})/2 \quad (2)$$

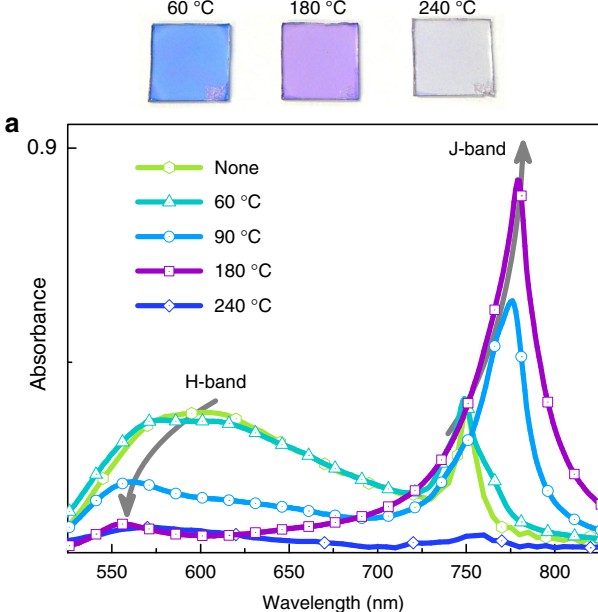

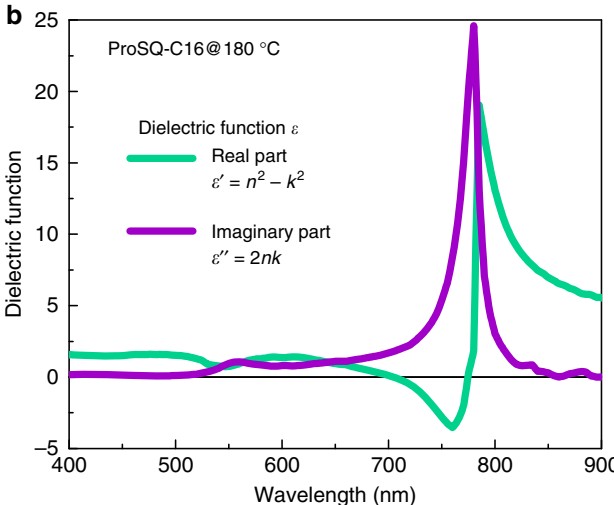

**Fig. 3** Unpolarized optical properties of ProSQ-C16 thin films. (**a**) Total absorbance spectra $-\log(T)$ of spin-casted ProSQ-C16 thin films on glass, with thicknesses around 25 nm, thermally annealed at the indicated temperatures. Annealing at 120 °C, 180 °C and 210 °C results in similar spectra, and for clarity of the presentation only the absorbance measured at 180 °C is shown. Gray arrows are to guide the eye to see the evolution of the H- and J-band with annealing temperature. Above the graph photographs of selected samples annealed at the indicated temperatures are shown. **b** Effective real and imaginary part of the dielectric function $\varepsilon' = n^2 - k^2$ and $\varepsilon'' = 2nk$, respectively, for ProSQ-C16 thin films annealed at 180 °C

and the ellipticity $\theta$ can be calculated according to:

$$\theta = \arctan\left(\frac{e^{CD} - 1}{e^{CD} + 1}\right) \qquad (3)$$

For CD values <0.5 rad the usual approximation for ellipticity:

$$\theta = CD/2 \qquad (4)$$

is accurate. While CD scales linear with pathlength (layer thickness) $\theta$ is only linear within this small value approximation, see Supplementary Fig. 6. Therefore, we chose in the following to

present our data in values of CD or CD per nm and not ellipticity (per nanometer).

The calculated differential Mueller matrices of typical samples annealed at 180 °C are presented in Fig. 4 for the (R,R)-enantiomer and in Fig. 5 for the (S,S)-enantiomer of ProSQ-C16. Their differential Mueller matrices agree well with the matrix symmetry in Eq. (1). The linear anisotropies of both samples are orders of magnitudes below the circular anisotropies, thus the anti-diagonal elements resemble the true, extraordinary large CB and CD of the samples. This is in coincidence with the microscopic inspections, cf. Fig. 2, not showing preferential in-plane ordering nor birefringence for the ProSQ-C16 thin films. Sample flipping to measure the Mueller matrix through the glass support results in congruent data sets, see Supplementary Fig. 7. Additionally, our thin film samples do not exhibit any measurable CP Bragg reflection, which has its origin in structural ordering[23,57]. This has been validated by performing Mueller matrix scans in reflection, Supplementary Fig. 8.

A strong, intrinsic CD signal is now associated with the J-band absorbance, while the weak H-band is CD-silent, see $l_{03}$ and $l_{30}$ within the differential Mueller matrices in Figs. 4 and 5. Interestingly, the line shape of the CD band is no longer proportional to the first derivative of the absorbance band, i.e. no clear bisignate couplet forms, even though the resonance is caused by strong exciton coupling[33,42,58]. However, the CD spectrum is still conservative, i.e. the spectral integral over the CD peak including the wider side bands approaches zero, see Supplementary Fig. 9.

The net Cotton effect is positive for the (R,R)-enantiomer meaning that it absorbs left circularly polarized light more strongly, while the net Cotton effect is negative for the (S,S)-enantiomer indicative for a preferential absorption of right circularly light. The CD-signal increases with increasing annealing temperature until it saturates and finally vanishes due to disruption of the thin film. The trend is depicted in Fig. 6a. While the J-band absorbance already saturates at 120 °C annealing, the long range coupling of the excitonic CD-signal further increases up to 180 °C annealing[37]. In Fig. 6b a collection of thickness normalized CD spectra of thin films annealed at the optimum temperature of 180 °C for both enantiomers are plotted. In addition to the experimental variance in peak maximum, we can see a trend that the sharp peak shifts from 780 nm to slightly longer wavelength with increasing layer thicknesses. The (R,R)-ProSQ-C16 performs better reaching (1000 ± 120) mdeg/nm while the maximum ellipticity of the (S,S)-ProSQ-C16 amounts only to (−650 ± 100) mdeg/nm under optimum annealing conditions. This is due to a lower enantiomeric purity resulting in inferior optical activity related performance most evident for the thin films. The (S,S)-enantiomer has an admixture of 2% of both the (R,R)-enantiomer and the (R,S)-diastereomer, which was evidenced by analytical chiral high-performance liquid chromatography (HPLC), see Supplementary Fig. 2 and explanation in Supplementary Note 2 for full details. While in solution the enantiomeric impurities will not reach the concentration threshold for self-aggregation, in the solid state they must somehow be integrated into the film at the expense of homochirality. All other (optical) properties, not connected to chirality, remain unaffected.

For an ultimate quantitative assessment of the optical activity, we calculate the dissymmetry factor $g$[6,22] for thin films annealed at 180 °C by normalizing the CD in units of absorbance $\Delta Abs^{CP}$ to the unpolarized absorbance of the sample. Note that we use $Abs = -\log(T)$ for the absorbance, while $A$ stands for the absorption. The differential circular absorbance of left (LCP) and right (RCP) CP light is calculated by:

$$\Delta Abs^{CP} = Abs^{LCP} - Abs^{RCP} = \frac{2}{\ln(10)} \cdot CD \qquad (5)$$

with CD according to Eq. (2).

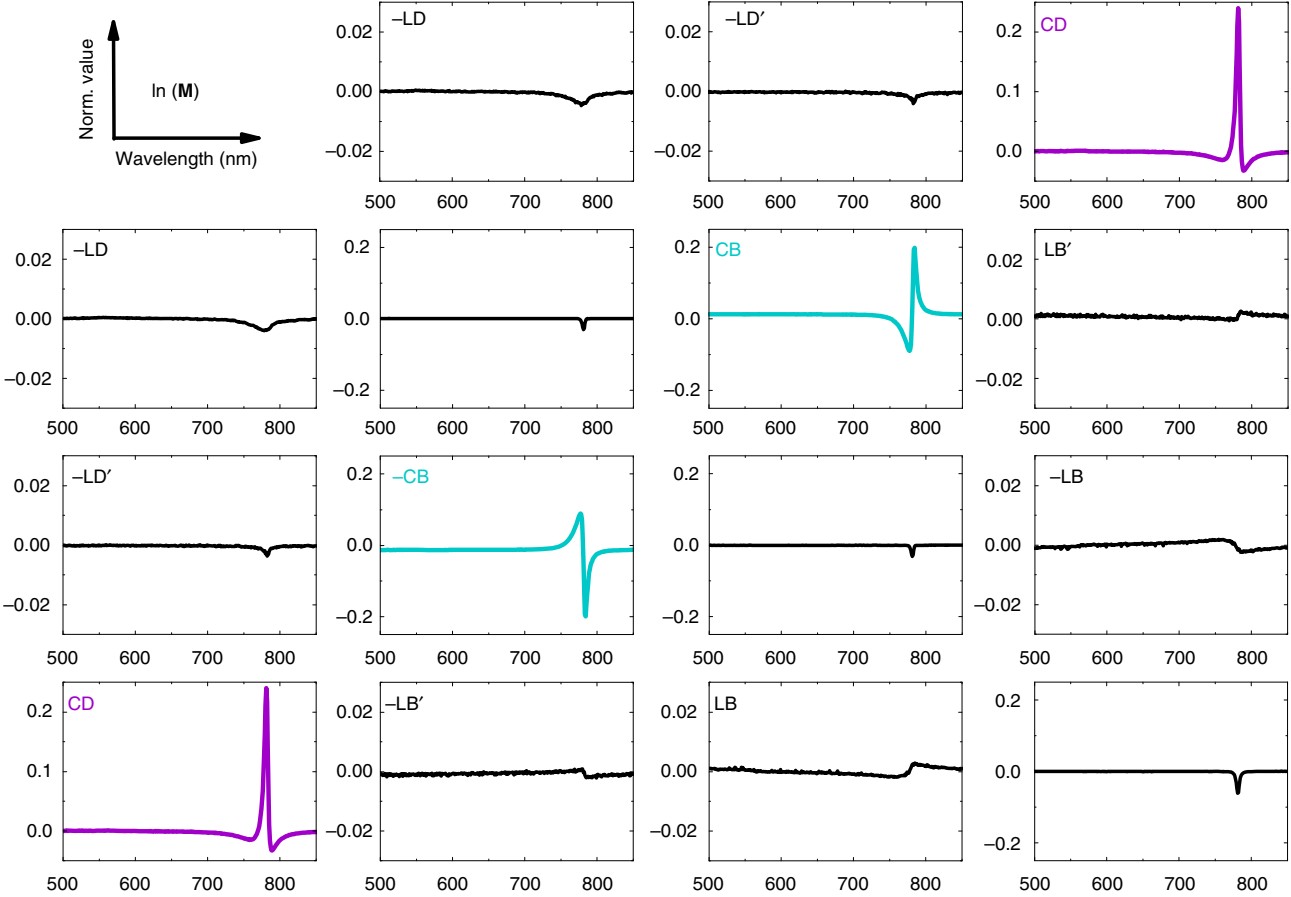

**Fig. 4** Mueller matrix spectroscopy. Calculated differential Mueller matrix **L** of ($R,R$)-ProSQ-C16 annealed at 180 °C with 16 nm film thickness. The assignment of the matrix elements is according to Eq. (1) in units of radian vs. wavelength in nm. The element $l_{00}$ equals zero and is omitted. Note that the scaling of the axes for the linear anisotropies is smaller by a factor of ten

The apparent dissymmetry factor $g_{ap}$ is obtained by:

$$g_{ap} = \frac{\Delta Abs^{CP}}{Abs^{meas}} \quad (6)$$

with $Abs^{meas}$ being the measured absorbance of the sample. For highly reflective thin film samples as the ProSQ-C16 samples, the measured absorbance must be corrected for reflection losses at the interfaces. We obtain this reflection corrected absorbance $Abs^{cor}$ for our ProSQ-C16 samples by a simple approach: The measured absorbance at its maximum at 780 nm is plotted against layer thickness $d$ and fitted linearly. The obtained slope is proportional to the absorption coefficient $\alpha$ according to:

$$-Abs^{meas} = -\log(1-R)^2 + \frac{\alpha}{\ln(10)} \cdot d. \quad (7)$$

where $(1-R)^2$ summarizes reflection losses at all interfaces for a free-standing film not accounting for coherent thin film interference[59]. To warrant this model to be sufficient, we provide further details in Supplementary Note 3 and Supplementary Fig. 10. The reflection-corrected absorbance $Abs^{cor}$ is then obtained by multiplying the slope of the linear fit with the layer thickness of interest. This value then replaces the measured absorbance in Eq. (6) to give the true dissymmetry factor $g_{true}$:

$$g_{true} = \frac{\Delta Abs^{CP}}{Abs^{cor}} \quad (8)$$

which is an intensive quantity, independent of the film thickness[22]. Note that the differential reflectivity of CP light is

zero under normal incidence for true = intrinsic chirality and the absence of CP Bragg reflection, thus $\Delta Abs^{CP}$ does not require any reflection correction[23].

The unpolarized maximum absorbance at 780 nm for 180 °C annealing is plotted in Fig. 6c for both enantiomers as a function of layer thickness. For determining the reflection corrected absorbance $Abs^{cor}$ according to Eq. (7) we fit only data points for an intermediate thickness range. A deviation from the simple model is noticeable for very thin layers (thinner than 5 nm) and thick layers (thicker than 50 nm). For thicknesses above 60 nm the films become opaque around 780 nm, see Supplementary Fig. 11 and are therefore not considered in our experiments.

In Fig. 6d the apparent and the true dissymmetry factors for both enantiomers for varying layer thicknesses are plotted. The data correspond to Fig. 6b. Thus, the peak maximum of $\Delta Abs^{CP}$ is subjected to the same spectral shift, however, for simplicity we speak about the spectral position being at around 780 nm in the following. Obviously, $g_{ap}$ scales with layer thickness (patterned pink squares and patterned light green circles, respectively), while $g_{true}$ (filled lilac squares and filled green circles, respectively) is thickness independent. Thus, our proposed reflection-correction remediates the artifactual thickness dependence of $g_{ap}$. When averaging over the data points for $g_{true}$ in Fig. 6d the true dissymmetry factor amounts to $0.75 \pm 0.07$ for the ($R,R$)-enantiomer (lilac filled squares) and to $-0.45 \pm 0.05$ for the ($S,S$)-enantiomer (green filled circles), respectively. This inferior performance of the ($S,S$)-enantiomer is consistent with thickness normalized ellipticity values. For intrinsic CD, the dissymmetry factor is expected to be independent of layer thickness. This is

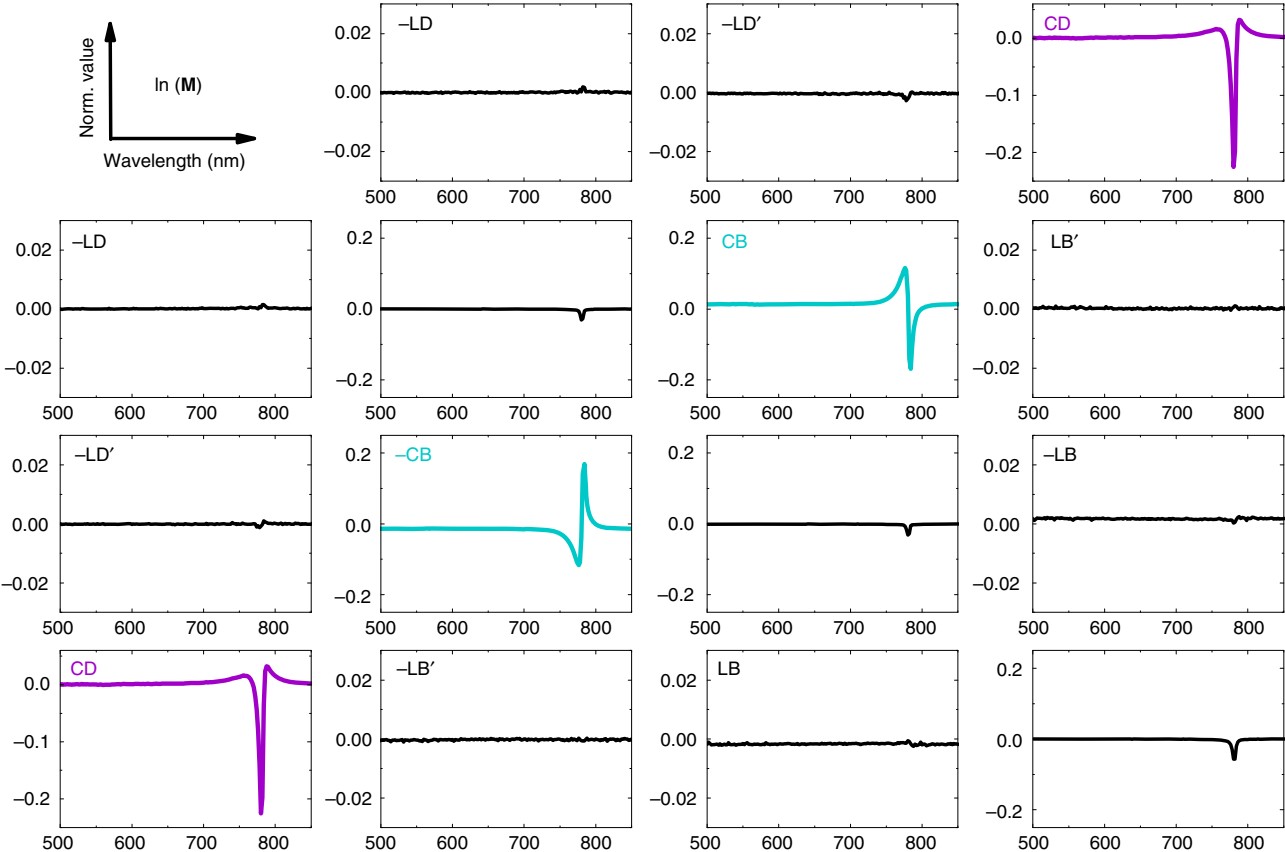

**Fig. 5** Mueller matrix spectroscopy. Calculated differential Mueller matrix (**L**) of (*S*,*S*)-ProSQ-C16 annealed at 180 °C with 20 nm film thickness. The assignment of the matrix elements is according to Eq. (1) in units of radian vs. wavelength in nm. The element $l_{00}$ equals zero and is omitted. Note that the scaling of the axes for the linear anisotropies is smaller by a factor of ten

actually the case within the experimental error. However, a linear fit through the data points suggests a slightly increasing $g_{true}$-value with layer thickness. Possibly, this can be attributed to simplicity of the model for obtaining the reflection-corrected absorbance and additional experimental uncertainties. To the best of our knowledge, a correction for reflection losses resulting in a larger true dissymmetry factor than the apparent has not been documented previously.

## Discussion

In the literature the *g*-value is commonly overestimated and needs to be corrected to smaller values to get the true *g*-value. This is because the measured CD signal is superimposed and artifactually increased by linear dichroism or amplified by mesoscopic structural ordering manifesting in thickness dependent circular Bragg reflection. The latter has been documented for spincasted and thermally annealed thin films of poly-paraphenylene-ethynylene[18] reaching a *g*-value of −0.38 and for poly-fluorenes even scoring −0.8[8] or −1[22]. These values are clearly identified by the authors as apparent dissymmetry factors due to mesoscopic structural ordering, which is absent in our ProSQ-C16 samples. The true dissymmetry factor reduces to $10^{-3}$ (note the change in sign) for the poly-fluorene samples[8]. Typical dissymmetry values for organic semiconductor thin films such as poly-[21,60] and oligo-thiophenes[20] range between $10^{-3}$ to $10^{-2}$. These are one to two orders of magnitude below the values of the annealed ProSQ-C16 thin films. For a detailed comparison we provide a table in the Supplementary information itemizing several recent publications on organic thin films, Supplementary Tab. 1.

However, these data are not acquired under the same conditions. Mostly they are measured by conventional CD spectrometers. Furthermore, no information about the dielectric function is given. Therefore, we characterized spincasted and thermally annealed thin films of a phenyl-bithiophene with chiral dimethylethylamine functional groups shortly named (*S*,*S*)-NPTTPN by Mueller matrix spectroscopy and ellipsometry in our labs. The molecular structural appearance of NPTTPN is similar to the ProSQ-C16 giving one reason for its selection. For details on the NPTTPN see Supplementary Note 6 and Supplementary Figs. 12 and 13. The NPTTPN thin films are free from mesoscopic structural ordering just as in case of the ProSQ-C16. As a marked difference the NPTTPN films show spectral signatures of weakly coupling H-aggregates, Supplementary Fig. 12c, instead of strongly coupling J-aggregates. However, both types are excitonic molecular aggregates, and the excitonic character of the optical response of the NPTTPN is also visible in the CD spectra, Supplementary Fig. 12e. The thickness normalized CD approximates to ±1 mdeg/nm and the apparent dissymmetry factor $g_{ap}$ to maximum values of $\pm 5 \times 10^{-3}$, Supplementary Fig. 12d and f. Thus, the excitonic origin of the CD signal alone in the absence of linear dichroism and Bragg reflection is not sufficient to boost the *g*-value. This typical excitonic CD signal of our NPTTPN thin films is by a factor of 150 smaller than the maximum CD values of our ProSQ-C16 samples. The weak excitonic coupling of the NPTTPN thin films is certainly not large enough to translate into a partially negative real permittivity, Supplementary Fig. 13, as it is the case for the ProSQ-C16 material. Therefore, no "excitonic amplification" of the CD can be expected for the NPTTPN. Thus, the exceptionalism of the ProSQ-C16 material lies in the fact of

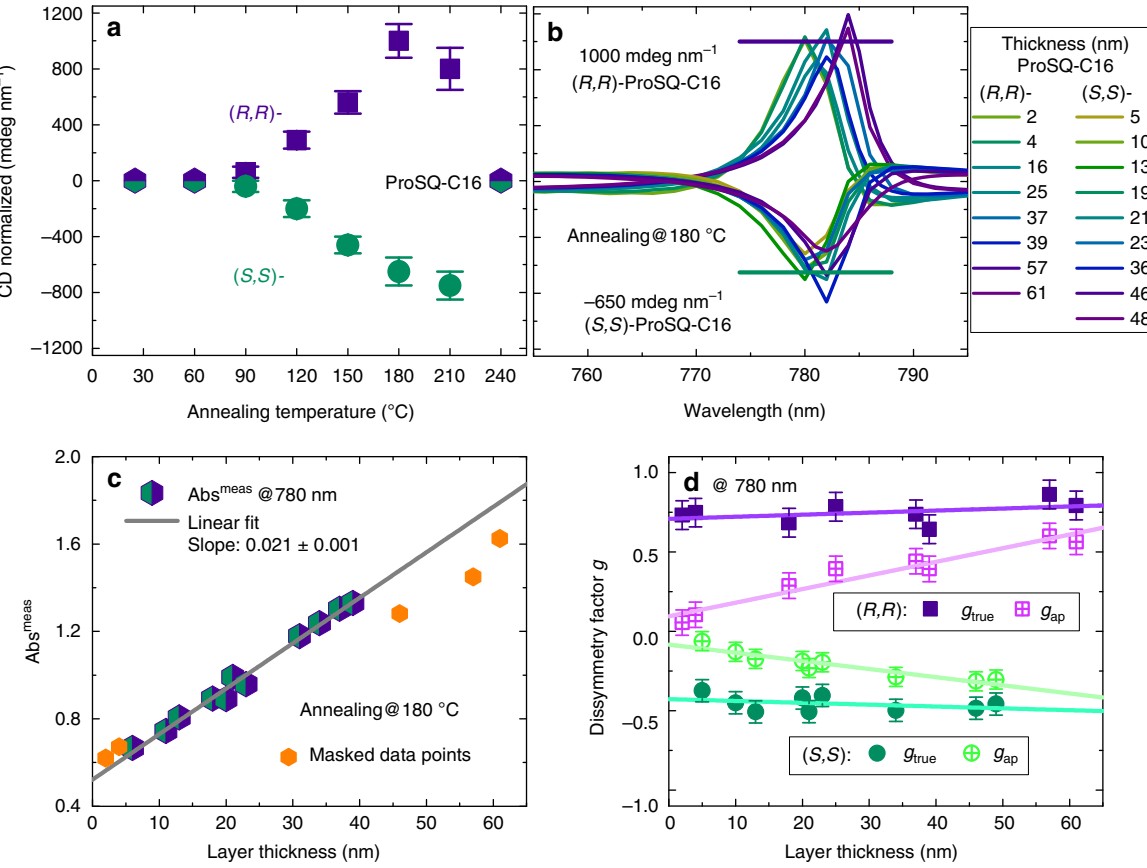

**Fig. 6** Thickness and absorbance normalized circular dichroic properties of ProSQ-C16 thin films. **a** Averaged maximum values of thickness normalized CD for varying annealing temperatures of (R,R)- and (S,S)-ProSQ-C16 thin films. The error bars are standard deviations. **b** Collection of thickness normalized CD spectra for thin films of varying thicknesses annealed at 180 °C. The layer thickness is color-coded as stated in the legend. **c** Unpolarized absorbance Abs$^{meas}$ vs. layer thickness. Data are peak values at around 780 nm from samples annealed at 180 °C. The unpolarized absorbance is valid for both enantiomers. From the linear fit of Abs according to Eq. (7) the reflection corrected absorbance Abs$^{cor}$ is calculated. **d** Maximum apparent and true dissymmetry factors $g_{ap}$ and $g_{true}$ according to Eqs. (6) and (8), respectively, at 780 nm against layer thickness for samples annealed at 180 °C. Straight lines in **c** and **d** are linear fits to the data points. Data in **b** and **d** are from the same samples and thus are convertible. Error bars are inferred from the experimental uncertainties of the measurements and are not standard deviations

showing a giant intrinsic CD not amplified by mesoscopic structural ordering but arising from the site of optical excitation. Here, the strong coupling of the molecular J-aggregates plays a crucial role, more precisely the large oscillator strength of the excitonic transition[42]. The real part of the dielectric function becomes negative over a small wavelength range just below the circular dichroic exciton resonance, see Fig. 3b. This feature is largely underexplored for organic matter even though it allows for resonant and bound electromagnetic modes within that range such as exciton-polaritons[48–51] possibly triggering novel concepts within the field of organic chiroptics. To conclude this work, we turned the spotlight on naturally homochiral squaraine based compounds and their ability to form strongly coupled J-aggregates to obtain an exceptionally high intrinsic CD in technologically applicable thin films. For that, we employ the powerful technique of Mueller matrix spectroscopy, to measure CD of thin films quantitatively and free from artifacts. Yet a correction for reflection losses due to the thin film nature is required to determine the true dissymmetry factor. A quantum chemical model is desirable for deeper understanding of the origin of the extraordinary strong CD. For now, we provide a new small molecular benchmark material for the development of organic thin film based chiroptics with a uniquely high CD rooted in a large oscillator strength of the J-type excitonic optical transitions and not in mesoscopic structural ordering.

## Methods

**Titration experiments**. Stock solutions with concentrations of $c = 0.175$ mM were prepared of both ProSQ-C16 enantiomers by dissolving 1.65 mg of the respective enantiomer in chloroform in 10 mL graduated flasks. For each measurement 0.1 mL of the respective stock solution were transferred into a 10 mm Hellma quartz cuvette and subsequently diluted with 3 mL of a chloroform-acetonitrile mixture to give a concentration of ~5.7 μM. For the solvent mixture the acetonitrile volume percentage is indicated ranging from 70% to 100% in steps of three. Cuvettes were gently shaken three times for equilibration and immediately measured. CD spectra were recorded with a Jasco J-810 spectro-polarimeter with 0.1 nm step size, 500 nm/min scan speeds, and 1 nm bandwidth. The instrument was calibrated with a reference sample in advance. UV–vis spectra were recorded on an Analytik Jena AG Specord 200 spectrometer with 0.2 nm step size.

**Thin film sample preparation**. (S,S)-ProSQ-C16 or (R,R)-ProSQ-C16 was dissolved in amylene stabilized chloroform (Aldrich, reagent grade), typically to give concentrations of 3 to 6 mg/mL. Thin films were prepared by spincoating for 60 s with varying rotation speeds and acceleration (1500 rpm, ramp 5; 3000 rpm, ramp 3; 4000 rpm, ramp 1) on 15 × 15 mm$^2$ glass substrates cut from VWR objective slides. Samples were subsequently thermally annealed for 90 min at the indicated temperatures. Preparation took place under inert atmosphere in a nitrogen filled glove box.

**Optical and atomic force microscopy**. Atomic force microscopy (AFM) images were recorded either in contact mode with an Agilent 5420 AFM equipped with a BudgetSensors Contact-G Al cantilever, or in intermittent contact mode with a JPK NanoWizard AFM (BudgetSensors Tap-300G, resonance frequency 300 kHz, force constant 40 N/m, tip radius <10 nm, and Nanosensors SSS-NCH, resonance

frequency 330 kHz, force constant 42 N/m, tip radius <5 nm) in combination with an inverted optical microscope (Nikon Eclipse TE 300) to image the very same sample region characterized by optical means before. Other optical microscopy images in darkfield mode or between crossed polarizers were obtained with an Olympus BX41 in reflection equipped with a DP12 digital camera. For bireflection microscopy, a Leica DMRME polarization microscope is used. Samples are rotated in between two crossed polarizers by a computer controlled rotation stage (see Supplementary Movies 1 and 2).

**Mueller matrix ellipsometry**. A Woollam variable angle spectroscopic ellipsometer (VASE) with a polarizer-compensator-sample-analyzer (PCSA) configuration and beam diameter of ~2 mm was used to determine optical constants, as well as to measure Mueller matrices, transmission spectra, and to determine layer thicknesses on the same spot on the sample by keeping the sample mounted in the same position in the sample holder.

Transmission data with linear s- and p-polarized light were measured in normal incidence and resulted in essentially the same transmission curves, which were therefore considered to resemble the unpolarized transmission. This was also validated by measurements in a conventional Cary 100 UV-Vis spectrometer with unpolarized beam. Absorbance equals $Abs = -\log(T)$. As expected for a sufficiently non-depolarizing sample, the global extinction is the same for polarized and unpolarized light.

Standard isotropic ellipsometry $\Psi$ and $\Delta$ data were acquired in reflection at varying angles of incidence ranging from 25° to 65° in 10° steps with spectral resolution of 5 nm. Data were fitted in the transparent infrared spectral regime from 1000 to 1700 nm by a Cauchy layer with the vendor's software WVASE32 to give the layer thickness, which was spot checked by AFM. Effective optical constants were obtained by extending the Cauchy model by a point-by-point fit over the complete spectral range. Normal incidence transmission data and s- and p-polarized reflection at 15° angle of incidence were included in the fitting. The MSE was smaller than 1, but the quality of the fit was ultimately assessed by how well the transmission and reflection data were reproduced. Either incoherent backside reflections from the glass substrate were considered in the fit routine, or the glass substrate backside was covered with milky Scotch tape during measurement.

We can record 11, i.e. the first three rows, out of the 15 normalized elements of the $(4 \times 4)$ Mueller matrix **M** with our single-compensator ellipsometer (PCSA configuration) by performing normal incidence transmission generalized ellipsometric scans. The Mueller matrix data are normalized to $m_{00}$. For determination of CD, scans were performed in normal incidence transmission with the monochromator entrance slit narrowed to 600 μm to reach a 2 nm spectral resolution. For cross validation purposes, additional Mueller matrix spectroscopy transmission measurements on a selected set of samples were done with a 4 photoelastic modulator (4-PEM) polarimeter[61]. This instruments measures all 15 elements of the normalized Mueller matrix simultaneously. Data comparison between both instruments was in good agreement and showed that, for these samples, we could deduct the missing matrix row by symmetry considerations. Finally, we can extract the CD spectrum from the differential Mueller matrix **L**, i.e. the matrix logarithm of **M**. The calculation is done in Matlab using the "logm" command. For a non-depolarizing system, the differential Mueller matrix the symmetry shown in Eq. (1). The element $l_{00}$ equals zero, because the original matrix **M** comes normalized to $m_{00}$. The other diagonal elements referring to the extinction properties of the sample can be affected by depolarization[55]. The anti-diagonal elements, resembling the circular anisotropic properties CB and CD of the sample, cannot be distorted by linear anisotropic effects, i.e. LD and LB, such as happens in conventional CD and CB spectroscopy methods[56]. We assess depolarization effects and prove them to be insignificant. For that, we use the Jones matrix quality factor $Q_{JM}$, which is output by the ellipsometer together with the measured 11 Mueller matrix elements[62]. Normalized $Q_{JM}$ spectra for all annealing temperatures are shown in Supplementary Fig. 14.

**Data availability**. The data that support the findings of this study are available from the corresponding author on reasonable request.

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

## Acknowledgements

We thank Prof. Dr. J. Parisi (head of Energy and Semiconductor Laboratory Division, University of Oldenburg) most sincerely for his shared-labs policy thereby providing access to a variety of state-of-the art equipment including technical support. We thank Andreas J. Schneider, University of Bonn, for conducting the analytical HPLC separation. The DFG-funded Research Training Group 1885 "Molecular Basis of Sensory Biology" is acknowledged for funding.

## Author contributions

M.Schulz did the synthesis of ProSQ-C16 and collected analytical data, summarized all analytical data, conducted the titration experiments, provided graphs for the manuscript, and searched the literature for benchmarking of the ellipticity. J.Z. contributed to synthesis, collection of analytical data and titration experiments. O.S.A. contributed to sample preparation and did the contact mode AFM measurements. S.B. did the synthesis of NPTTPN and collected and summarized all analytical data. F.B. did the intermittent contact mode AFM measurements, polarized microscopy, calculation of optical spectra, typesetting of manuscript, was involved in data discussion, and contributed to manuscript writing. A.L. supervised M.Schulz and J.Z. and did proofreading of the manuscript. O.A. conducted the 4-PEM Mueller matrix spectroscopy, provided the concept of reflection-corrected absorbance, was deeply involved in data discussion and contributed to manuscript writing. M.Schiek initiated the experiments, prepared samples, did all single-compensator Mueller matrix spectroscopy and transmission measurements and all data analysis thereof, measured and fitted the spectroscopic ellipsometry data, recorded (polarized) optical microscopy images, prepared all figures, and wrote the manuscript.

## Additional information

**Competing interests:** The authors declare no competing interests.

