## [Peer Review File · Nature Communications]

Reviewers' comments:

Reviewer #1 (Remarks to the Author):

The article "Giant Intrinsic Circular Dichroism of Prolinol-Derived Squaraine Thin Films" describes thorough spectroscopic investigations of the optical activity of a material whose "strong circular dichroism" and preparation was already described in reference 9. Therefore the novelty is not extremely high, and it is rather an extension of that work.

The work seems thoroughly done, and the paper is well written, excepting its tendency to overstatement and extrapolation.

To describe the observed effect as "revolutionary" is not scientifically sound. It is claimed that there is no documented rival of this compound as a supramolecularly optically active material, but supramolecular or indeed molecular systems are not compared, and no other examples are given. This should have been done.

The "ex-chiral pool strategy" is one used by virtually all researchers in the area of chiral supramolecular materials, to write as if the lead group have invented it seems absurd, and in any case this aspect is quite irrelevant for the present paper.

A detail that needs addressing is that in Figure 6b, the thicknesses should be indicated, as there is some spread in the position of the band.

Also, in the Introduction, it is stated that CD machines are "well designed for diluted solutions and isotropic colloidal suspension(s)", in fact they work well for all isotropic materials that don't scatter light of the wavelengths at which they are studied.

If this work really does "hands on a plate for exploring excitons-polaritons" that is what should be done, rather than simply "rediscover(ing)" phenomena that have already been published (quotes are from the paper). Something which could have added weight was the calculation of the origins of the claimed effect. Only in these ways can a communication status be warranted, which takes nothing away from the high quality of the research which is appropriate as a full paper in a specialist journal as it is.

Reviewer #2 (Remarks to the Author):

This paper presents and discusses results with a potential very high impact and it will very likely deserve publication in Nature Communications.

Apart from the extraordinary magnitude of the CD effects, I appreciated very much the accuracy and thoroughness of the optical analysis. Foremost there is the use of the complete Mueller matrix analysis for thin films chiroptical measurements. But I found extremely interesting the SI section, dedicated to correcting absorption (A) for the reflection (R) contribution and this is indeed where I would like to know more. Figure S4 shows that the contribution of R is extremely large, to the point that most of the observed loss would actually be due to this term rather than to A. Following the authors' analysis, for a 10 nm thickness $A=0.21$, instead of about $A = 0.75$, one would naively measure. I believe this correction is absolutely correct, but to my knowledge no one in this context has ever performed it before. Accordingly, the literature data may generally suffer from a massive underestimation of the so-called g-factor, possibly by very large factors (in this case one would estimate around 3.5). If so, the value $g=0.75$ reported here should not be compared with something around 0.1, which is nowadays a recurrent figure in organized thin films of organic molecules or polymers, but rather with 0.35 or possibly even more, depending on R (or other losses due e.g. to scattering) in other films. This does not withdraw interest to the paper, but it should be said openly. To this end, the correction applied must be (briefly) described in the main text, rather than be somewhat demoted in the SI. At the same time, in order to be fair with preceding literature, the apparent value of g found here should be reported, i.e. the -value one

would obtain using total absorbance rather than the R-corrected A. A line mentioning this should appear also in the abstract

Going more into details of the SI, I do not understand why the black line of Figure 5 does not match the red line of Figure 4 (I thought they should be identical). How it comes that Eq. 4 with several free parameters gives such a poor fit to the experimental points. Finally, what is indeed the gain of using more complex (= with more variables) models if the one represented by Eq. 3 looks quite satisfactory? Let me also comment that green and red curves of Figure 5 both suggest a trend, which does not at all seem to fit the data. Altogether, does this simply mean that interference does not play a role? Please explain better.

I should expect a remarkable difference in the contribution of reflection, depending on the geometry of the experiment: if light enters the thin film from air or from the solid substrate, where the film is deposited. If I am not wrong, this should be revealed by a very simple experiment consisting in flipping the same sample when running the measurements. Please either provide an explanation that this hypothesis is inconsistent or do run this experiment.

More in general, the SI lacks several experimental details, concerning e.g. the synthetic procedure, the instrumentation used for all measurements (including XRD, thickness, dielectric function, ...), the supports where the film is deposited, the conditions for coating (solvent, speed, acceleration). As a further and final point, the authors must be well aware that for large values of dichroism, the relationship between ellipticity and dichroism fails. They should explain if and how did they take care of this problem.

Reviewer #3 (Remarks to the Author):

This paper reports the synthesis and materials science of enantiomers of a squaraine compound "ProSQ-C16", ultimately using ellipsometry to determine its "true", and apparently large, circular dichroism. It should be emphasised that the synthesis, aggregation properties and optoelectronic properties of the compound under study are well known by the authors, with all the associated data on one of the enantiomers recently reported (ref 9, Schulz et al, Phys.Chem.Chem.Phys.,2017, 19, 6996). Indeed, there is heavy repetition in this manuscript. Thus, it would seem there are only two aspects of this submission that could qualify it as sufficiently novel for Nat Comm: Either the general use of ellipsometry to characterise the mechanisms that underpin observed chiroptical responses in thin films of chiral materials, or the large value of the true CD obtained for ProSQ-C16. In terms of the former, a cursory literature survey suggests there to be limited use of ellipsometry to study chiral materials, however there is precedent (for example ACS Nano, 2017, 11, 12713). Therefore, it would seem the usage of the technique in and of itself is not sufficiently novel. In terms of the large value of true CD obtained, the authors claim that the value "has no documented rival". The validity of this statement rather depends on how many systems have been measured. Is there no rival because only the measurements of a couple of systems have been documented? Or is the key value measured (i.e. mdeg/nm) material specific and therefore comparisons are irrelevant? The authors only appear to cross reference the value obtained for ProSQ-C16 against a bilayer of graphene; a material which seems to be a poor comparison (from a materials perspective).

In summary, the sole aspect of novelty in this paper appears to be the high true CD, which is not sufficient to justify publication in Nat Comm. If the authors were able to better benchmark the value obtained for their materials against a range of other aggregating supramolecular systems (either using literature values, if available, or determining them experimentally), then this paper could be reconsidered.

Other comments:

Written Style: In general, the language is too verbose for Communications paper to a broad readership. phrases like "highly optical pure films", are not well defined. Statements such as "This

hands on a plate for exploring exciton-polaritons and other ultra-strong light matter interactions, which will be part of our future endeavors”, “Prospectively, a (quantum mechanical) structural model” make the text and any scientific discussion particularly difficult to follow. The term ‘quasimetallic’ is peppered throughout the abstract and conclusion, without any discussion of what it means in this context.

Aggregation data: The text states that the lack of vibronic structure in the CD spectra implies a π -stacking distance is 3.5 Å; whilst Spano et al (ref 33) have calculated this explicitly for lutein aggregates and it is unclear that it would still hold for colloidal squaraine. The only clear conclusion from a lack of vibronic structure is a strong exciton coupling, where excitations are transferred much faster than the period of vibration. Further information about the aggregation in solution could be obtained by variable temperature measurements (at fixed 100%ACN, for instance) and Dynamic Light Scattering (to compare the aggregates size with the solid state).

Thin film data: Regarding the thin films, there is no discussion about why the temperature range 60 – 180 was chosen (other than because these were used in ref 9 and ref 34) nor accompanying cross-polarised microscope images to confirm the lack of morphological features. If the AFM images are recorded at the same temperature, this should be provided, as should a comparison with the same films at room temperature. If they are recorded at different temperatures, it is difficult to compare the two images provided in Figure 2 (a and b) as they are at different scales. If the statement “the patterns around the crystallites in crossed polarized optical microscopy images originate from a slightly improved order” is referring to Figure 2 (c), this is unclear, and perhaps a more zoomed in image would be more useful. What is the order improved with respect to, other areas on the film or a pristine (non-annealed) sample? Given the thermal annealing is so crucial for the H-aggregate formation, a more local characterisation of the aggregate structure would be interesting. For example, this could be achieved using Resonant Raman Spectroscopy, which has been used before to understand squaraine aggregate species. The microscope images reveal significant variations over the surface of the film at the microscale (Fig (c)) – it is hard to visualise how the ‘same spots’ are selected on the sample for all spectroscopic measurements.

Differences between enantiomers: The explanation for the lower enantiomeric purity of (S,S) is supported by HPLC data. However, in solution (Figure 1) the difference is not noticeable. Therefore, there must be some non-linear amplification of chirality in thin films which should be investigated more deeply.

Reviewer #1

The article "Giant Intrinsic Circular Dichroism of Prolinol-Derived Squaraine Thin Films" describes thorough spectroscopic investigations of the optical activity of a material whose "strong circular dichroism" and preparation was already described in reference 9. Therefore the novelty is not extremely high, and it is rather an extension of that work.

The work seems thoroughly done, and the paper is well written, excepting its tendency to overstatement and extrapolation.

To describe the observed effect as "revolutionary" is not scientifically sound. It is claimed that there is no documented rival of this compound as a supramolecularly optically active material, but supramolecular or indeed molecular systems are not compared, and no other examples are given. This should have been done.

We have cleared up the text from overstatements and buzzwords, e.g., we have replaced "revolutionary" and "no documented rival" by "new benchmark" in the abstract as well as in the main text.

We have thoroughly searched current literature about CD of organic thin films and present a detailed summary in form of a table in the Supplementary Information for proper benchmarking of our material. Actually, a comparison is sometimes not so straightforward, since systematic quantitative investigations accounting for possible artifacts on circular dichroism are rather rare. As a whole, in all the reported organic thin films the dissymmetry value downscales to 10^{-2} - 10^{-4} and the ellipticity to being below 10 mdeg/nm when subtracting amplification effects due to mesoscopic structural ordering.

The "ex-chiral pool strategy" is one used by virtually all researchers in the area of chiral supramolecular materials, to write as if the lead group have invented it seems absurd, and in any case this aspect is quite irrelevant for the present paper.

It was not our intention to claim that we have invented the ex-chiral-pool strategy, and we apologize for leaving this impression. We no longer mention the strategy in the abstract and have moved all details regarding the synthesis to the Supplementary Information. We also decided to provide all synthetic details, even though it is largely redundant to our previous publication, according the inquiry of Reviewer #2 asking for even more details on the synthetic procedure.

A detail that needs addressing is that in Figure 6b, the thicknesses should be indicated, as there is some spread in the position of the band.

We have completely revised Fig. 6: We extended the annealing temperature range (see also Reviewer #3) and added normalized CD data to Fig. 6(a) for non-annealed samples, and annealing at 60°C, 210°C and 240°C. Thereby we show, that 180°C-annealing is the optimum annealing temperature and continue the detailed discussion for this annealing. In Fig. 6(b) the layer thickness is now color-coded. Indeed, there is a trend that for thicker layers the peak maximum slightly shifts to longer wavelengths, which we also denote in the manuscript text. Furthermore, we provide the not-normalized CD in units of absorbance together with the conventional absorbance in Fig. 6(c). We added the apparent g-values (not using the reflection-corrected absorbance but the apparent) to the true g-values (reflection-corrected absorbance) for all layer thicknesses, which is now Fig. 6(d).

Also, in the Introduction, it is stated that CD machines are "well designed for diluted solutions and isotropic colloidal suspension(s)", in fact they work well for all isotropic materials that don't scatter light of the wavelengths at which they are studied.

We have changed the sentence in the introduction commenting about the use of commercial CD-spectrometers and added two references: "Pseudo-CD effects are concomitantly probed for (anisotropic) thin film samples due to the oversimplified utilization of (commercial) CD-spectropolarimeters, which are well-designed for diluted solutions and any other isotropic samples, but not for the quantitative assessment of circular dichroism of typical thin film samples.\cite{Castiglioni2009, Harada2013}"

If this work really does "hands on a plate for exploring excitons-polaritons" that is what should be done, rather than simply "rediscover(ing)" phenomena that have already been published (quotes are from the paper).

We have deleted these phrases from the text.

Something which could have added weight was the calculation of the origins of the claimed effect.

Indeed, quantum mechanical calculations providing a sound structural model and a quantitative explanation of the origin of the circular dichroism are highly desirable for us. But conducting such theoretical studies and calculations would need the development of a suitable theoretical approach first, which would definitely go beyond the scope of this work. In fact, we have just started collaborating with Prof. Stefan Grimme, University of Bonn, on those calculations. The outcome will take time, since the calculations are actually very elaborate and require a specialized analysis. However, we think that the record-high values of CD of our organic thin film samples are well supported by our experimental approach. Therefore, we think that the experimental description we provide of the phenomenon (thin films fabrication and accurate optical characterization) is worth to be published as a Communication.

Only in these ways can a communication status be warranted, which takes nothing away from the high quality of the research which is appropriate as a full paper in a specialist journal as it is.

Circular dichroism of organic matter was already discovered in the mid-nineteenth century. Due to its tiny-quantity-effect, it found its application as spectroscopic tool, nowadays widely applied though, for structure resolving purposes. With the advent of metamaterials and their macroscopically measurable pseudo CD, circular dichroism has come into focus for device application purposes, see e.g. the Special Issue No. 16 "Chirality and Nanophotonics" in Adv. Opt. Mater. 5 (2017). Organic materials have not been seriously considered for this, yet, due to the tininess of CD. The present manuscript opens the door for further research on up to now overlooked large-quantity CD of organic matter. Our investigated material is, currently a unique, benchmark material and can serve as a guiding model compound for further development of chiral cyanine-like J-aggregates with large absorbance oscillator strength. Thereby, supramolecular circular dichroism escapes from its niche and becomes highly relevant for a broad readership, justifying publication in *Nature Communications*.

Reviewer #2

This paper presents and discusses results with a potential very high impact and it will very likely deserve publication in Nature Communications.

Apart from the extraordinary magnitude of the CD effects, I appreciated very much the accuracy and thoroughness of the optical analysis. Foremost there is the use of the complete Mueller matrix analysis for thin films chiroptical measurements. But I found extremely interesting the SI section, dedicated to correcting absorption (A) for the reflection (R) contribution and this is indeed where I would like to know more. Figure S4 shows that the contribution of R is extremely large, to the point that most of the observed loss would actually be due to this term rather than to A. Following the authors' analysis, for a 10 nm thickness $A=0.21$, instead of about $A = 0.75$, one would naively measure. I believe this correction is absolutely correct, but to my knowledge no one in this context has ever performed it before.

We agree, also to the best of our knowledge such reflection correction has not been presented before.

Accordingly, the literature data may generally suffer from a massive underestimation of the so-called g -factor, possibly by very large factors (in this case one would estimate around 3.5). If so, the value $g=0.75$ reported here should not be compared with something around 0.1, which is nowadays a recurrent figure in organized thin films of organic molecules or polymers, but rather with 0.35 or possibly even more, depending on R (or other losses due e.g. to scattering) in other films.

We agree the reflection correction is very important in our kind of thin film samples and, because it is not considered in other literature works, they may show an underestimated g -value. But we have also found some works that suffer from an overestimation of g -values due to structural ordering (see table S1 in the Supplementary Information). If liquid crystalline ordering is present, CD also manifests in reflection (Bragg-reflection) thereby contributing to the overall transmission-measured CD (in this case arising to the differential reflectivity of CPL and not from differential absorption). For our material we could exclude Bragg-reflection, see also the new figure S8 in the Supplementary Information, so we are confident that the g -values we report are realistic.

This does not withdraw interest to the paper, but it should be said openly. To this end, the correction applied must be (briefly) described in the main text, rather than be somewhat demoted in the SI.

We moved a description of the reflection correction into the main text. The linearly fitted absorbance data can now be seen in Fig. 6(c).

At the same time, in order to be fair with preceding literature, the apparent value of g found here should be reported, i.e. the $-$ value one would obtain using total absorbance rather than the R-corrected A.

It was not our intention to be unfair or to inflate our data. We present the uncorrected, not-normalized CD-values in units of absorbance in Fig. 6(c). Absorbance-normalized CD-values, using

both as-measured absorbance and reflection-corrected absorbance, are displayed in Fig. 6(d). It is obvious, that the *apparent g*-value is depending on layer thickness, which is not in accordance with the intensive character of a correct *g*-value. If we pick a peak value of *g* for a 60 nm thick film of the R-enantiomer, we obtain a value of around 0.6. Thus, also the apparent *g* gives an estimate of the extraordinary strength of the present circular dichroism, and scores off structurally amplified CD data from the literature.

A line mentioning this should appear also in the abstract.

We added “ ... after accounting for reflection losses related to the thin film nature,...” to the abstract.

Going more into details of the SI, I do not understand why the black line of Figure 5 does not match the red line of Figure 4 (I thought they should be identical). How it comes that Eq. 4 with several free parameters gives such a poor fit to the experimental points. Finally, what is indeed the gain of using more complex (= with more variables) models if the one represented by Eq. 3 looks quite satisfactory? Let me also comment that green and red curves of Figure 5 both suggest a trend, which does not at all seem to fit the data. Altogether, does this simply mean that interference does not play a role? Please explain better.

The data presented in former SI-Fig.5, now SI-Fig.10, are based on calculations using the optical constants (see Fig.3) of the thin films obtained from fitting of ellipsometric data. These data are for a free-standing film, while measured absorbance data contain the contribution of the substrate. Thus, the measured absorbance (grey dots in SI-Fig.10) is offset from calculations on free-standing films to lower values by a constant value, but the *slope* (the relevant parameter for the reflection correction) is the same for both cases. When adding film-glass and glass-air interfaces (green curve), the curve is offset to lower values accordingly.

We wanted to make sure, that even the simplest fit is valid, so that there is no substantial difference in applying more complex optical models. Note also that even the more elaborated coherent model of a film on a substrate does not provide a fully satisfactory fit in SI-Fig.10. More refinement could be possible since the calculations there do not account for surface roughness but assume infinite smoothness, which is not true: R_{rms} is about 6 nm of the real films. However, according to our wave-optic calculations, consideration of a more elaborated wave-optics model for the thin film substrate system does not offer any substantial difference in the final reflection corrected *g*-values for the normal incidence configuration and in the spectral region of interest. Note that only in a configuration different from the normal-incidence illumination a more elaborated optical model becomes really necessary.

I should expect a remarkable difference in the contribution of reflection, depending on the geometry of the experiment: if light enters the thin film from air or from the solid substrate, where the film is deposited. If I am not wrong, this should be revealed by a very simple experiment consisting in flipping the same sample when running the measurements. Please either provide an explanation that this hypothesis is inconsistent or do run this experiment.

We provide the requested sample flipping experiment in the Supplementary Information in SI-Fig.7: CD and absorbance are the same no matter if the sample is measured from front- or backside under normal incidence. This is an expected result since (true) CD and absorbance are not expected to change when reversing the wavevector. Note also that, while the presence of the substrate can make that both configurations have a different reflectance, the overall Fresnel reflection losses suffered by the transmitted beam are the same regardless of which side of the sample is illuminated first. Thus, our approach to reflection corrected absorbance is mostly adequate for both measurement geometries.

More in general, the SI lacks several experimental details, concerning e.g. the synthetic procedure, the instrumentation used for all measurements (including XRD, thickness, dielectric function, ...), the supports where the film is deposited, the conditions for coating (solvent, speed, acceleration).

When experimental details are not given in the Supplementary Information, they can be found in the Methods section of the main paper, e.g. conditions for coating and ellipsometry. Regarding the XRD, we have included another measurement and more information on the experimental data. Regarding the synthesis, we moved all information into the Supplementary Information due to the redundancy to our previous publication as also pointed out by Reviewer #1. We hope that this is a satisfying compromise.

As a further and final point, the authors must be well aware that for large values of dichroism, the relationship between ellipticity and dichroism fails. They should explain if and how did they take care of this problem.

Yes, this is true, the linear approximation ellipticity = CD/2 is only a small value approximation, and indeed our data exceed the range valid for the linear approximation, see SI-Fig.6. Therefore, we refrain from providing ellipticity but present CD and thickness normalized CD instead.

Reviewer #3

This paper reports the synthesis and materials science of enantiomers of a squaraine compound "ProSQ-C16", ultimately using ellipsometry to determine its "true", and apparently large, circular dichroism. It should be emphasised that the synthesis, aggregation properties and optoelectronic properties of the compound under study are well known by the authors, with all the associated data on one of the enantiomers recently reported (ref 9, Schulz et al, Phys.Chem.Chem.Phys.,2017, 19, 6996). Indeed, there is heavy repetition in this manuscript.

We agree with the reviewer that most information regarding the synthesis has already been presented in our PCCP paper, ref. 9. Therefore, we have moved all information on the synthesis into the Supplementary Information, see also the comments for Reviewers #1 and #2.

Thus, it would seem there are only two aspects of this submission that could qualify it as sufficiently novel for Nat Comm: Either the general use of ellipsometry to characterise the mechanisms that

underpin observed chiroptical responses in thin films of chiral materials, or the large value of the true CD obtained for ProSQ-C16. In terms of the former, a cursory literature survey suggests there to be limited use of ellipsometry to study chiral materials, however there is precedent (for example ACS Nano, 2017, 11, 12713).

The mentioned article has been published on the web after we had submitted our manuscript. We thank the reviewer for bringing this publication to our attention are glad to cite this article and use its findings for benchmarking purposes.

Therefore, it would seem the usage of the technique in and of itself is not sufficiently novel. In terms of the large value of true CD obtained, the authors claim that the value “has no documented rival”.

We have replaced this expression by “new benchmark material”.

The validity of this statement rather depends on how many systems have been measured. Is there no rival because only the measurements of a couple of systems have been documented? Or is the key value measured (i.e. mdeg/nm) material specific and therefore comparisons are irrelevant? The authors only appear to cross reference the value obtained for ProSQ-C16 against a bilayer of graphene; a material which seems to be a poor comparison (from a materials perspective).

In summary, the sole aspect of novelty in this paper appears to be the high true CD, which is not sufficient to justify publication in Nat Comm. If the authors were able to better benchmark the value obtained for their materials against a range of other aggregating supramolecular systems (either using literature values, if available, or determining them experimentally), then this paper could be reconsidered.

We totally agree with the reviewer that the main aspect of novelty is the high value of true CD of the squaraine thin films and tried to make it clearer in the text. We have added an extended table to the Supplementary Information summarizing recent literature on CD in organic thin film samples. Actually, a comparison is sometimes not so straightforward, since systematic quantitative investigations accounting for possible artifacts on circular dichroism are rather rare. In our literature research we have found that for typical organic thin films a dissymmetry value downscales to 10^{-2} - 10^{-4} and the ellipticity to being below 10 mdeg/nm when subtracting amplification effects due to mesoscopic structural ordering. See also comments for Reviewers #1 and #2. However, we think that there are other organic materials, e.g., chirally J-aggregated cyanine dyes with strong absorbance properties, showing a similar CD effect, they just have not been investigated in that sense. Therefore, we hope that our paper will increase attention for this class of possible new and efficient materials for organic thin film chiroptical devices.

Written Style: In general, the language is too verbose for Communications paper to a broad readership. phrases like “highly optical pure films”, are not well defined. Statements such as “This hands on a plate for exploring exciton-polaritons and other ultra-strong light matter interactions, which will be part of our future endeavors”, “Prospectively, a (quantum mechanical) structural model” make the text and any scientific discussion particularly difficult to follow. The term ‘quasimetallic’ is peppered throughout the abstract and conclusion, without any discussion of what it means in this context.

We have changed the term “highly optical pure films” into “highly enantiopure films”.

We have replaced the colloquial term “quasimetallic reflection” in the abstract by the scientifically sound term “a negative real part of the dielectric function”.

We no longer extrapolate on potential future research in the sense of exciton-polaritons or ultra-strong light-matter interaction, thus, we have deleted the text passages. The conclusion has been completely rewritten to take into account the reviewer’s criticisms.

Aggregation data: The text states that the lack of vibronic structure in the CD spectra implies a π -stacking distance is 3.5 Å; whilst Spano et al (ref 33) have calculated this explicitly for lutein aggregates and it is unclear that it would still hold for colloidal squaraine. The only clear conclusion from a lack of vibronic structure is a strong exciton coupling, where excitations are transferred much faster than the period of vibration.

We refrain from making an assumption of the pi-stacking distance based on the work by Spano et al for lutein aggregates and have rephrased the sentence to a simple half-sentence “... which is indicative for a strong excitonic coupling.\cite{Spano2009}”

Further information about the aggregation in solution could be obtained by variable temperature measurements (at fixed 100%ACN, for instance) and Dynamic Light Scattering (to compare the aggregates size with the solid state).

Since the colloidal aggregation in solution is not in focus of the present paper, we have not conducted temperature dependent solution-based aggregation studies. We think that this might overload the manuscript, which is supposed to be a communication, with sideshow information. Regarding the suggested DLS investigation, we are not able to provide reasonable data: Our DLS-system Zetasizer NanoZS operates with a laser at 633nm, which is a typical wavelength for such instruments. The measurement is based on scattering, and the laser light must not be absorbed by the particles of interest. Thus, our material of interest is incompatible with the instrument.

Thin film data: Regarding the thin films, there is no discussion about why the temperature range 60 – 180 was chosen (other than because these were used in ref 9 and ref 34)

We have extended the presented annealing temperature range of our samples which includes now no annealing, 210°C and 240°C annealing. We have inserted the corresponding data into Fig. 3(a) showing the total absorbance and into Fig. 6(a) showing thickness normalized CD. Thereby we show, that the shape of absorbance spectra changes from no annealing to 120°C annealing, while it remains constant between 120°C and 210°C, and breaks up for annealing at 240°C due to dewetting (maybe even decomposition) of the organic matter. The CD continues to increase, while absorbance has already saturated, up to its maximum for 180°C annealing. CD remains more or less constant for 210°C annealing and collapses together with all other thin film properties at 240°C annealing. Thereby, we cover the complete dynamic range of the sample system including also its boundaries, and the 180°C annealing emerges as optimum when aiming at maximum circular dichroic properties.

nor accompanying cross-polarised microscope images to confirm the lack of morphological features.

In Fig.2(d) a cross-polarized microscopy image is shown, and in Fig.2(c) a corresponding dark-field microscopy images. Images are overexposed for better visibility, demonstrating the lack of morphological features with the exception of the described contaminations by rather large grains. We have added movies pieced together from series of bireflectance microscopy images of samples rotated between crossed polarizers to the Supplementary information for further illustration. The Mueller matrix data in Figs 4 and 5 confirm that the values in the differential Mueller matrix with the physical meaning LD, LD', LB and LB' (linear dichroism/birefringence properties, visible in bireflectance microscopy) are orders of magnitudes smaller than the circular dichroic properties.

If the AFM images are recorded at the same temperature, this should be provided, as should a comparison with the same films at room temperature. If they are recorded at different temperatures, it is difficult to compare the two images provided in Figure 2 (a and b) as they are at different scales.

The thin film morphology of all samples with no annealing up to 210°C annealing look alike, i.e., they all show the characteristic random morphology with the 3 nm step heights and a few randomly distributed clusters as shown in Fig. 2. Fig. 2(a) shows a larger area of the film also capturing a characteristic cluster of the material, while Fig. 2(b) is a close up of the sample. We have changed the misleading term “crystallite” into “cluster”, since the clusters are not necessarily composed of crystallized material. We apologize for our previous misleading naming. Such clusters are randomly distributed over all samples at all annealing temperatures. Images exemplarily shown here are recorded on samples annealed at 180°C. Various samples were quickly scanned by contact mode AFM, images are not shown due to inferior quality, selected samples were then scanned by intermittent contact mode AFM to give images in publication quality. Note that for annealing at 240°C the film properties are destroyed, AFM and microscopy images are shown in the Supplementary information Fig. S2.

While morphology is very robust against thermal annealing, the optical properties, i.e., CD- and absorbance-spectra, severely depend on the annealing temperature. Therefore, we take great care of connecting optical data to the annealing temperature of the respective samples throughout the paper.

If the statement “the patterns around the crystallites in crossed polarized optical microscopy images originate from a slightly improved order” is referring to Figure 2 (c), this is unclear, and perhaps a more zoomed in image would be more useful. What is the order improved with respect to, other areas on the film or a pristine (non-annealed) sample?

A more zoomed in image is provided, now Fig.2(d), between crossed polarizers exhibiting a faint “birefringence pattern” around the cluster. Movies put together from a series of microscopy images between rotated crossed polarizers are added to the Supplementary information for clarity. The inferior quality of the movies is connected to the really low birefringence/bireflectance of the samples. The “bireflectance pattern” stem from an increased order just around the cluster, which does not translate into a change in morphology. This is the same for all annealing temperatures. However, we refrain from saying that this is due to increased order, since there is still no crystallographic long-range order visible by X-ray diffraction, to avoid confusion.

Given the thermal annealing is so crucial for the H-aggregate formation, a more local characterisation of the aggregate structure would be interesting. For example, this could be achieved using Resonant Raman Spectroscopy, which has been used before to understand squaraine aggregate species.

The clusters can by now means assigned to be H-aggregated species. The thin films are clearly dominated by the optical signatures of J-aggregated species. Unfortunately, we do not have access to a resonant Raman microscopy to provide localized Raman spectroscopy of the samples. Raman is indeed a powerful tool, but adding a Raman investigation to the manuscript would clearly go beyond the scope of a Communication.

The microscope images reveal significant variations over the surface of the film at the microscale (Fig (c)) – it is hard to visualise how the ‘same spots’ are selected on the sample for all spectroscopic measurements.

The illuminating spot for recording the Mueller matrix and transmission data as well as reflection and ellipsometry data has a diameter of about 2 mm. It averages thereby over a lot of the randomly distributed clusters. They do not affect the measurement. The increased but still faint birefringence around the clusters is spurious for the circular dichroic properties, which are orders of magnitudes stronger. For this see the complete Mueller matrix in Figs 4 and 5. The sample is fixed to the sample holder during all measurements. Thereby we can safely say, that we illuminate the same spot of 2 mm in diameter for CD, absorbance and ellipsometry (for layer thickness) measurements. Note that the microscopy images zoom into these areas and provide only views of micrometer-sized areas. We have added the following in the Methods section: “...by keeping the sample in mounted in the same position in the sample holder. Note that measurements average over a macroscopic sample area due to the 2 mm spot diameter of the illuminating light.”

Differences between enantiomers: The explanation for the lower enantiomeric purity of (S,S) is supported by HPLC data. However, in solution (Figure 1) the difference is not noticeable. Therefore, there must be some non-linear amplification of chirality in thin films which should be investigated more deeply.

The minor enantiomeric impurities of the (S,S)-enantiomer is very unlikely to translate into solution-based aggregation studies: For the aggregation preferably enantiomers of same handedness will assemble, c.f. chiral discrimination. The concentration of minor enantiomeric impurities, e.g., the here present 2%, will be too low to overcome the threshold for colloidal precipitation. Since the monomers are CD-silent, they will not be noticeable within the CD-spectra. In thin films, on the contrary, the impurities are just there and in the same phase, which is the solid state, and no concentration driven equilibrium between dissolved and colloidal form is present. Therefore, the thin film samples are by far less tolerant for impurities, and they will easily impair performance.

We have added to the text (titration experiment discussion):

“Within the range of experimental error the CD-spectra of both colloidal enantiomer solutions are mirror-imaged but otherwise equivalent. Any minor enantiomeric impurities (see also Supplementary information) will not be noticeable, since only molecules of the same handedness assemble, and the

concentration threshold for aggregation of the undesired enantiomer or diastereomer is not reached.”

and (thin film discussion):

“While in solution the enantiomeric impurities will not reach the concentration threshold for self-aggregation, in the solid state they must somehow be integrated into the lm at the expense of homochirality. All other (optical) properties, not connected to chirality, remain unaffected.”

Reviewers' comments:

Reviewer #1 (Remarks to the Author):

The authors have replied thoroughly and respectfully to the comments and criticisms provided to them by all three referees.

However, the main concerns identified by referees 1 and 3 have not been remedied, and therefore the paper seems unsuitable for publication in this journal. Apart from comments made previously, that are largely still valid, I would add the following arising from the replies:

The comparison with other materials is partially provided, but this list is not exhaustive and doesn't help tremendously. The present work would be aided enormously by showing just one other type of material under the same conditions.

The synthesis has been included, but it is irrelevant (already reported).

The dependence of optical activity on thickness is confirmed, but not explained, and calls into question the validity of the reported results, something also evident in the fact that values for (S,S) and (R,R) enantiomers are very different in their values (they should be exactly opposite). The argument from the authors about non-linear effects is not scholarly.

Reviewer #2 (Remarks to the Author):

The ms. has undergone quite a complete revision and it now reads much better. My own concerns were addressed, but some new ones emerged.

The origin of the CD spectrum in the thin films cannot be assigned to exciton coupling, or at least this mechanism cannot be dominant. If it were so, the integral about the absorption band would be 0, which is clearly not the case. The expression "it is of excitonically coupled nature" sounds very funny (it should read "it is due to exciton coupling") but it is wrong: whichever sum of bisignate couplet would result in CD bands of alternating signs, and not in a very prominent one of one sign. For this reason, I should reject all the interpretation at the edge between pages 11 and 12. While the CD of the aggregates in solution is clearly suggestive of exciton coupling and I agree with the view of tilted H-stacks shown in Figure 1 and in the text, the extraordinarily large CD of the film remains of unknown origin.

I should push this argument forward. Because the CD spectrum suggests the need for a different type of interaction, I cannot fully support the view that J-aggregates are responsible for the appearance of the absorption spectrum either.

In conclusion, I still believe that this is a relevant paper for its findings and for the methods and I still support publication in Nat Comm, but only after speculations are reduced and CD interpretation is corrected (or erased, without loss of interest).

As a further definitely minor comment to the rebuttal. I am not convinced of the statement: "the overall Fresnel reflection losses suffered by the transmitted beam are the same regardless of which side of the sample is illuminated first", because to my (weak) understanding it depends on the difference in refractive indices: on the two faces of the sample, light enters the glass of the support from air or from the film and the two situations look very different to me. By far I am not expert in this field and for this reason, I more seek enlightening than criticize the authors' work.

Reviewer #3 (Remarks to the Author):

The paper has been considerably improved since the first submission, and the authors have responded well to the comments made. It remains that the main novelty of the study lies in the use of Mueller Matrix spectroscopy for determination of true Circular Dichroism. It should be noted that there are other studies using this approach that are not cited/referred to by the authors (e.g. Arturo Mendoza-Galván et al 2018 J. Opt. 20 024001).

Technical comments:

Figure 6c is overly chaotic and should be improved.

The data for non-thickness normalized CD (Fig 6 (a) and 6(b)) should be provided in the Supporting Information.

I do not understand why the dissymmetry factor varies with layer thickness (the authors state this is obvious). Replicate values and associated error bars should be added to check this trend is indeed 'within experimental error', as stated.

**Rebuttal letter to initiate appeal for decision on manuscript NCOMMS-17-33733-A
“Giant Intrinsic Circular Dichroism of Prolinol-Derived Squaraine Thin Films”**

We thank the referees for their critical reading of our manuscript and for their valuable remarks which helped us to clarify and improve some important aspects. We are convinced that we have aided the main concern: the lack of a proper benchmarking for the extraordinary high circular dichroism, by providing the full characterization of another material under the same conditions. For this, we include the work of another co-author. Since we believe that in some points the reviewers are mistaken or might have misunderstood points we wanted to make, we have tried to make parts of the paper more clear. The major changes/additions are copied into this letter. In the following we will not only present the advances to our manuscript but provide a point-by-point response (green text) to the reviewer comments (printed in italics colored red).

Reviewer #1

The authors have replied thoroughly and respectfully to the comments and criticisms provided to them by all three referees.

However, the main concerns identified by referees 1 and 3 have not been remedied, and therefore the paper seems unsuitable for publication in this journal. Apart from comments made previously, that are largely still valid, I would add the following arising from the replies:

The comparison with other materials is partially provided, but this list is not exhaustive and doesn't help tremendously. The present work would be aided enormously by showing just one other type of material under the same conditions.

We are convinced that we can provide this now, and have added the following to the Supplementary information, page 18 ff (Note that the citation of references in this response letter does not match the numbering in the main paper or Supplementary information):

Benchmarking: (S,S)-NPTTPN

Figure S12 (a) Structural formula of **(S,S)-NPTTPN**. Characterization of **(S,S)-NPTTPN** thin films annealed at 60°C with varying layer thicknesses (86 nm and 128 nm): (b) Overexposed microscopy image between crossed polarizers. (c) Absorbance spectra. (d) Thickness normalized circular dichroism spectra. (e) Thickness normalized CD (black curve) and derivative of absorbance spectrum (red curve). (f) *Apparent* dissymmetry factor $g_{(ap)}$.

The phenyl-bithiophene with chiral dimethylethylamine functional groups (1*S*,1'*S*)-1,1'-([2,2'-bithiophene]-5,5'-diylbis(4,1-phenylene))bis(*N,N'*-dimethylethylamine) shortly named **(S,S)-NPTTPN** has been synthesized by us as outlined below. It is used as reference material for circular dichroism in thin film samples, which were prepared and measured under the same conditions as the **ProSQ-C16** samples, see Methods in the main paper. The structural formula of **(S,S)-NPTTPN** is shown in Fig. S12 (a). It is similar to the **ProSQ-C16** in the sense that it is a rod-like molecular semiconductor with an amine-based terminal homochiral functional group being on both ends of the molecule. Upon spincoating of **(S,S)-NPTTPN** from a chloroform solution and subsequent thermal annealing a featureless film is formed with faint birefringence. Thus, no amplification of the CD is expected due to mesoscopic structural ordering. An overexposed microscopy image between crossed polarizers after annealing at 60°C is displayed in (b). Note that higher annealing temperatures did not induce crystallization, and neither enhanced the circular dichroism, therefore we show the best data obtained for the 60°C annealing.

The absorbance spectra in (c) of **NPTTPN** films with varying layer thicknesses show typical spectral signatures of weakly coupled molecular H-aggregates with pronounced vibronic transitions. [Spano2010] These vibronic progressions are also noticeable in the circular dichroism spectra displayed in (d). The CD spectrum is a superposition of bisignate bands for each vibronic progression, which has also been documented for polythiophene thin films. [Langeveld-Voss2000] The CD spectrum has a bisignate appearance with a positive lobe at shorter wavelength and a negative lobe at longer wavelength and the maximum CD values approximate to 1 mdeg/nm. The CD spectrum is proportional to the derivative of the absorbance spectrum, see (e), clearly indicating an excitonic CD signal. Finally, in (f) the absorbance normalized *apparent* dissymmetry factor $g_{(ap)}$ is plotted revealing maximum values of $5 \cdot 10^{-3}$ which is a typical value for polythiophene [Langeveld-Voss2000, Lakhwani2007], poly-(phenylene)-bithiophene [Koeckelberghs2007] and oligothiophene [Albano2017] thin films, see also Table S1 above. Thus, the excitonic origin of the CD signal alone is not sufficient to boost the g -value.

We refrain from calculating the *true* g -value, $g_{(true)}$ since we would need additional samples to perform the reflection-correction. More substantially, due to the vibronic progressions, we would need to refine the concept of our proposed reflection-correction, which goes beyond the scope of the benchmarking. Note that our molecule of main interest, the **ProSQ-C16**, does not show vibronic progressions due to strong intermolecular interactions. However, we clearly demonstrate by the analysis of **(S,S)-NPTTPN** that typical excitonic CD signals are by a factor of 150 smaller than the maximum CD values of our new benchmark material **ProSQ-C16**.

Effective optical constants have been determined by spectroscopic ellipsometry as described in the Methods section of the main paper. Here in addition to the described procedure, surface roughness needed to be taken into account by an effective medium layer with 50% voids. In Fig. S13 (a) the optical constants and in Fig. S13 (b) the real and imaginary part of the dielectric function of a **NPTTPN** thin film annealed at 60°C are displayed. The values of the optical constants are typical for an organic semiconductor but rather on the lower edge. [Vezie2016] The oscillator strength is certainly not large enough to translate into a negative real part of the dielectric function as it is the case for the **ProSQ-C16** material. Thus, no "excitonic amplification" of the circular dichroism of the **NPTTPN** can be expected.

Figure S13 (a) Effective optical constants and (b) real and imaginary part of the dielectric function of a **NPTTPN** thin film.

Synthesis and Analytical Details of (S,S)-NPTTPN

A 50 mL two-necked flask equipped with a condenser was charged with 272 mg (0.66 mmol) of 5,5'-diiodo-2,2'-bithiophene, 400 mg (1.45 mmol, 2.2 equiv.) of (S)-1-(4-N,N-dimethylethylamine)phenylboronic acid pinacol ester, 840 mg (3.98 mmol, 6 equiv.) of potassium phosphate, 60 mg (0.066 mmol, 10 mol%) of tris(dibenzylideneacetone)dipalladium(o) (Pd2(dba)3), and 91 mg (0.165 mmol, 25 mol%) of 1,1'-bis(diphenylphosphino)ferrocene (dppf) under an argon atmosphere. After adding 14 mL 1,4-dioxane and 3.5 mL water the reaction mixture was degassed and heated to 100°C for 42 h under an argon atmosphere. The water phase was extracted three times with dichloromethane, and the combined organic phase was dried over sodium sulphate. The solvent was removed under reduced pressure and the remaining crude product was purified by column chromatography on silica gel using cyclohexane:ethyl acetate (1:5) plus 5% triethyl amine as eluent, R_f factor: 0.25.

Yield: 125mg (60 %)

Molecular formula: $C_{28}H_{32}N_2S_2$

Molar mass: 460.70 g/mol

Optical rotation: $[\alpha]_D^{20} = -99.0^\circ$ in dichloromethane 0.5 mg/mL

UV/Vis: $[\lambda]_{\max} = 378$ nm in dichloromethane

Fluorescence: $[\lambda]_{\max} = 460$ nm in dichloromethane

ESI+ HRMS (Bruker microTOF-Q): m/z calcd for $C_{28}H_{32}N_2S_2H$ $[M+H]^+$: 461.2080, found: 461.2080.

Elemental analysis (Heraeus Vario EL): $C_{28}H_{32}N_2S_2 \cdot CH_2Cl_2$ calcd C: 71.93, H: 6.92, N: 5.97, S: 13.67; found: C: 72.12, H: 6.58, N: 5.88, S: 13.46.

1H NMR (Bruker DPX 400): (400 MHz, CD_2Cl_2 , RT) [ppm] = 7.57 (d, $J = 8.2$ Hz, 4 H, H-6), 7.34 (d, $J = 8.2$ Hz, 4 H, H-5), 7.26 (d, $J = 3.8$ Hz, 2 H, H-9), 7.19 (d, $J = 3.8$ Hz, 2 H, H-10), 3.26 (q, $J = 6.7$ Hz, 2 H, H-3), 2.18 (2, 12 H, H-1), 1.34 (d, $J = 6.7$ Hz, 6 H, H-2).

^{13}C NMR (Bruker DPX 400): (100 MHz, CD_2Cl_2 , RT) [ppm] = 145.1 (C-4), 143.6 (C-7), 136.9 (C-11), 133.0 (C-8), 128.7 (C-5), 125.9 (C-6), 125.0 (C-10), 124.1 (C-9), 66.0 (C-3), 43.5 (C-1), 20.4 (C-2).

[Spano2010] Spano, F. C. The Spectral Signatures of Frenkel Polarons in H- and J-Aggregates. *Acc. Chem. Res.* 2010, 43, 429-439.

[Langeveld-Voss2000] Langeveld-Voss, B.; Janssen, R.; Meijer, E. On the origin of optical activity in polythiophenes. *J. Mol. Struct.* 2000, 521, 285-301.

[Lakhwani2007] Lakhwani, G.; Koeckelberghs, G.; Meskers, S. C. J.; Janssen, R. The chiroptical properties of chiral substituted poly [3-((3S)-3,7-dimethyloctyl)thiophene] as a function of film thickness. *Chem. Phys. Lett.* 2007, 437, 193-197.

[Koeckelberghs2007] Koeckelberghs, G.; Vangheluwe, M.; Persoons, A.; Verbiest, T. Chirality in Poly(phenylene-alt-bithiophene)s: A Comprehensive Study of Their Behavior in Film and Nonsolvents. *Macromolecules* 2007, 40, 8142-8150.

[Albano2017] Albano, G.; Lissia, M.; Pescitelli, G.; Aronica, L. A.; Bari, L. D. Chiroptical response inversion upon sample flipping in thin films of a chiral benzo[1,2-b:4,5-b']dithiophenebased oligothiophene. *Mater. Chem. Front.* 2017, 1, 2047-2056.

[Vezie2016] Vezie, M. S.; Few, S.; Meager, I.; Pieridou, G.; Döring, B.; Ashraf, R. S.; Goni, A. R.; Bronstein, H.; McCulloch, I.; Hayes, S. C. et al. Exploring the origin of high optical absorption in conjugated polymers. *Nat. Mater.* 2016, 15, 746.

The discussion in the main paper (page 15) has been changed to:

More commonly, the measured CD signal is superimposed and artifactually increased by linear dichroism or amplified by mesoscopic structural ordering manifesting in thickness dependent circular Bragg reflection. The latter has been documented for spincasted and thermally annealed thin films of poly-paraphenylene-ethynylene [Wilson2002] reaching a g -value of -0.38 and for poly-fluorenes even scoring -0.8 [DiNuzzo2017] or -1. [Lakhwani2011] These values are clearly identified by the authors as *apparent* dissymmetry factors due to mesoscopic structural ordering, which is absent in our **ProSQ-C16** samples. The *true* dissymmetry factor reduces to 10^{-3} (note the change in sign) for the poly-fluorene samples. [DiNuzzo2017] Typical dissymmetry values for organic semiconductor thin films such as poly-[Lakhwani2007, Langeveld-Voss2000] and oligo-thiophenes [Albano2017] range between 10^{-3} to 10^{-2} . These are one to two orders of magnitude below the values of the annealed **ProSQ-C16** thin films. For a detailed comparison we provide a table in the Supplementary information itemizing several recent publications on organic thin films, SI-Tab. S1.

However, these data are not acquired under the same conditions. Mostly they are measured by conventional CD spectrometers. Furthermore, no information about the dielectric function is given. Therefore, we characterized spincasted and thermally annealed thin films of a phenyl-bithiophene with chiral dimethylethylamine functional groups shortly named **(S,S)-NPTTPN** by Mueller matrix spectroscopy and ellipsometry in our labs, for details see Supplementary Information SI-Fig.s S12 and S13. The **NPTTPN** thin films are free from mesoscopic structural ordering and show spectral signatures of weakly coupling H-aggregates. The excitonic character of the optical response is also visible in the CD spectra. The thickness normalized CD approximates to 1 mdeg/nm and the *apparent* dissymmetry factor $g_{(ap)}$ to maximum values of $5 \cdot 10^{-3}$. Thus, the excitonic origin of the CD signal alone in the absence of linear dichroism and Bragg reflection is not sufficient to boost the g -value. This typical excitonic CD signal of our **NPTTPN** thin films is by a factor of 150 smaller than the maximum CD values of our **ProSQ-C16** samples. The weak excitonic coupling of the **NPTTPN** thin films is certainly not large enough to translate into a partially negative real permittivity as it is the case for the **ProSQ-C16** material. Thus, no "excitonic amplification" of the circular dichroism can be expected for the **NPTTPN**.

Thus, the exceptionalism of the **ProSQ-C16** material lies in the fact of showing a giant *intrinsic* CD not amplified by mesoscopic structural ordering but arising from the site of optical excitation. Here, the strong coupling of the molecular J-aggregates plays a crucial role, more precisely the large oscillator strength of the excitonic transition. [Markova2013] The real part of the dielectric function becomes negative over a small wavelength range just below the circular dichroic exciton resonance, see Fig. 3(b). This allows for resonant and bound electromagnetic modes within that range such as exciton-polaritons [Tristani-Kendra1984,Gu2013,Gentile2014,Woo2017] possibly triggering novel concepts within the field of organic chiroptics.

[Wilson2002] J. N. Wilson, W. Steffen, T. G. McKenzie, G. Lieser, M. Oda, D. Neher, and U. H. F. Bunz, Chiroptical properties of poly (*p*-phenyleneethynylene) copolymers in thin films: large g -values," J. Am. Chem. Soc. 124, 6830-6831 (2002).

[DiNuzzo2017] D. Di Nuzzo, C. Kulkarni, B. Zhao, E. Smolinsky, F. Tassinari, S. C. Meskers, R. Naaman, E. Meijer, and R. H. Friend, High circular polarization of electroluminescence achieved via self-assembly of a light-emitting chiral conjugated polymer into multidomain cholesteric films, ACS Nano 11, 12713-12722 (2017).

[Lakhwani2011] G. Lakhwani and S. C. J. Meskers, Insights from chiral polyfluorene on the unification of molecular exciton and cholesteric liquid crystal theories for chiroptical phenomena, J. Phys. Chem. A 116, 1121-1128 (2011).

[Lakhwani2007] G. Lakhwani, G. Koeckelberghs, S. C. J. Meskers, and R. Janssen, The chiroptical properties of chiral substituted poly [3-((3s)-3,7-dimethyloctyl)thiophene] as a function of film thickness, Chem. Phys. Lett. 437, 193-197 (2007).

[Langeveld-Voss2000] B. Langeveld-Voss, R. Janssen, and E. Meijer, On the origin of optical activity in polythiophenes, J. Mol. Struct. 521, 285-301 (2000).

[Albano2017] G. Albano, M. Lissia, G. Pescitelli, L. A. Aronica, and L. D. Bari, Chiroptical response inversion upon sample flipping in thin films of a chiral benzo[1,2-b:4,5-b']dithiophene based oligothiophene, Mater. Chem. Front. 1, 2047-2056 (2017).

[Markova2013] L. I. Markova, V. L. Malinovskii, L. D. Patsenker, and R. Häner, J- vs. H-type assembly: pentamethine cyanine (Cy5) as a near-IR chiroptical reporter, Chem. Commun. 49, 5298-5300 (2013).

[Tristani-Kendra1984] M. Tristani-Kendra and C. J. Eckhardt, Influence of crystal fields on the quasimetallic reflection spectra of crystals: Optical spectra of polymorphs of a squarylium dye, J. Chem. Phys. 81, 1160-1173 (1984).

[Gu2013] L. Gu, J. Livenere, G. Zhu, E. E. Narimanov, and M. Noginov, Quest for organic plasmonics, Appl. Phys. Lett. 103, 021104 (2013).

[Gentile2014] M. J. Gentile, S. Núñez-Sánchez, and W. L. Barnes, Optical field-enhancement and subwavelength field-confinement using excitonic nanostructures, Nano Lett. 14, 2339-2344 (2014).

[Woo2017] B. H. Woo, I. C. Seo, E. Lee, S. Y. Kim, T. Y. Kim, S. C. Lim, H. Y. Jeong, C. K. Hwangbo, and Y. C. Jun, Dispersion control of excitonic thin films for tailored superabsorption in the visible region, ACS Photonics 4, 1138-1145 (2017).

The synthesis has been included, but it is irrelevant (already reported).

The synthesis for the (*S,S*)-enantiomer has been reported previously, yes, but it is important to note that we have slightly modified the synthetic route to improve its purity, and that we also obtained the (*R,R*)-enantiomer following this protocol. Additionally, we provide chiral HPLC analysis of the isomers' enantiomeric purity, which is essential to answer the next point. We value the transparency of the experimental procedure very high, and have understood that *Nature Communications* likewise does, so we think that all experimental data, also the synthetic procedure in brief, should be provided in the Supplementary Information (we do not wish to present synthetic details in the main paper). Thereby, we also try to make a compromise to the request of Reviewer#2 from the previous report to add even more details on the synthetic procedure.

*The dependence of optical activity on thickness is confirmed, but not explained, and calls into question the validity of the reported results, something also evident in the fact that values for (*S,S*) and (*R,R*) enantiomers are very different in their values (they should be exactly opposite). The argument from the authors about non-linear effects is not scholarly.*

The reason why the values of (*S,S*)- and (*R,R*)-enantiomer are not the same (but opposite in sign) can be understood from the synthetic procedure, which relies on the enantiomeric purity of the chiral pool starting material. The chiral HPLC analysis reveals admixture of the other enantiomer and/or diastereomers for the (*S,S*)-enantiomer, which translates into a reduced circular dichroism of the *S,S*-enantiomer processed to thin films but not for the solution experiments. We *never* came up with the argument of non-linear effects. This argument was brought up by Reviewer#3 in the first review. Indeed, 2% impurity does not mean 2% less circular dichroism, but a more severe reduction in performance. This is comparable to doping: addition of small quantity dopant induces a large macroscopic effect, which can, naively speaking, be called a non-linear effect. However, since we cannot provide a conclusive structural model, we refrain from quantifying the impurity effect. Let us repeat the comment of the referee (italics) and the explanation we gave, which still holds true:

*Differences between enantiomers: The explanation for the lower enantiomeric purity of (*S,S*) is supported by HPLC data. However, in solution (Figure 1) the difference is not noticeable. Therefore, there must be some non-linear amplification of chirality in thin films which should be investigated more deeply.*

The minor enantiomeric impurities of the (*S,S*)-enantiomer is very unlikely to translate into solution-based aggregation studies: For the aggregation preferably enantiomers of same handedness will assemble, c.f. chiral discrimination. The concentration of minor enantiomeric impurities, e.g., the here present 2%, will be too low to overcome the threshold for colloidal precipitation. Since the monomers are CD-silent, they will not be noticeable within the CD-spectra. In thin films, on the contrary, the impurities are just there and in the same phase, which is the solid state, and no concentration driven

equilibrium between dissolved and colloidal form is present. Therefore, the thin film samples are by far less tolerant for impurities, and they will easily impair performance.

We have added to the text (titration experiment discussion):

“Within the range of experimental error the CD-spectra of both colloidal enantiomer solutions are mirror-imaged but otherwise equivalent. Any minor enantiomeric impurities (see also Supplementary information) will not be noticeable, since only molecules of the same handedness assemble, and the concentration threshold for aggregation of the undesired enantiomer or diastereomer is not reached.”

and (thin film discussion):

“While in solution the enantiomeric impurities will not reach the concentration threshold for self-aggregation, in the solid state they must somehow be integrated into the film at the expense of homochirality. All other (optical) properties, not connected to chirality, remain unaffected.”

Regarding the thickness dependence of “optical activity” we need to differentiate: optical activity subsumes different quantities such as circular dichroism, ellipticity, circular birefringence, optical rotation and dissymmetry factor (absorbance normalized CD in “units of absorbance”). Circular dichroism (CD) can be extracted from the differential Mueller matrix as briefly outlined in the paper. CD depends linear on layer thickness d according to: $CD = (2\pi/\lambda) \cdot (k_L - k_R) \cdot d$ with λ being the wavelength and k_L and k_R the extinction coefficients for left and right circularly polarized light. On the contrary, the g -value (absorbance normalized CD in “units of absorbance”) is an intensive quantity and does not depend on layer thickness. We think that we give the discussion on the required reflection-correction to obtain the true thickness-independent g -value for our thin films quite some room in the paper. Actually, the necessity of this correction is one of the conclusions of the paper.

The coauthor Oriol Arteaga, is highly experienced in Mueller matrix spectroscopy and has published numerous articles on the theory of Mueller matrix spectroscopy used for determination of CD, and we refer to a few of these articles to provide the interested reader with in-depth information. We do not think that it is required in the present paper to repeat well reported theory since here we have a clear focus on material characterization.

Reviewer #2

The ms. has undergone quite a complete revision and it now reads much better. My own concerns were addressed, but some new ones emerged.

The origin of the CD spectrum in the thin films cannot be assigned to exciton coupling, or at least this mechanism cannot be dominant. If it were so, the integral about the absorption band would be 0, which is clearly not the case.

Actually, the integral approaches zero when taking the wider side bands into account. The following figure has been added to the Supplementary information:

Figure S9: Circular dichroism versus photon energy (black line) of an S,S-ProSQ-C16 sample annealed at 180°C and the spectral integral (cyan line/area) over the signal. The integration starts at 1.4 eV and the cyan curve displays the mathematical area underneath the CD curve (black line). The cyan shading is for illustration purposes. When the upper bound of integration reaches 1.9 eV the integral becomes zero. Thus, when the wider side peaks are included the spectral integral approaches zero indicative for an excitonic CD response.

The expression “it is of excitonically coupled nature” sounds very funny (it should read “it is due to exciton coupling”) but it is wrong: whichever sum of bisignate couplet would result in CD bands of alternating signs, and not in a very prominent one of one sign. For this reason, I should reject all the interpretation at the edge between pages 11 and 12. While the CD of the aggregates in solution is clearly suggestive of exciton coupling and I agree with the view of tilted H-stacks shown in Figure 1 and in the text, the extraordinarily large CD of the film remains of unknown origin. I should push this argument forward. Because the CD spectrum suggests the need for a different type of interaction, I cannot fully support the view that J-aggregates are responsible for the appearance of the absorption spectrum either.

We do not have a complete picture of the origin of the large CD of the film and we agree that the expression “it is of excitonically coupled nature” was not well written and have therefore rephrased it:

“Interestingly, the line shape of the CD band is no longer proportional to the first derivative of the absorbance band, i.e. no clear bisignate couplet forms, even though the resonance is caused by strong exciton coupling. [Spano2009, Kobayashi2010, Markova2013] However, the CD spectrum is still conservative, i.e. the spectral integral over the CD peak including the wider side bands approaches zero, see Figure S9.”

Thus, the spectral integral over the CD bands goes approximately to zero. There is a large CD peak, but it is very sharp and narrow, and its area is compensated by the wider side peaks of opposite sign. This type of spectral response suggests dipolar transition moments in an oblique configuration with respect to the plane of the sample. In these conditions, there may be two different exciton coupling contributions to the overall experimental absorption: one for the components parallel to the plane of the sample and the other for the perpendicular components. Each one of them can give rise to a

bisignate Cotton effect of opposite sign with respect to the other. To qualitatively illustrate this behavior, we have plotted the superimposition (sum) of two slightly different Lorentz bisignated oscillators with opposite Cotton sign (out of phase) in Figures A and B. They lead to spectral shapes similar to those observed in our CD measurements. Needless to say, that the simulated CD spectra are conservative, and the integral of all the spectra shown is zero. We are not in a position yet to give a quantitative or more detailed explanation about the origin of the CD spectrum but, contrary to what it is suggested by the referee, we think that strong exciton coupling plays an important role and it cannot be ruled out.

additional Figure A

additional Figure B

In conclusion, I still believe that this is a relevant paper for its findings and for the methods and I still support publication in Nat Comm, but only after speculations are reduced and CD interpretation is corrected (or erased, without loss of interest).

We have erased the speculation from the manuscript on the line shape of the CD-spectrum. But we have further emphasized the strong excitonic coupling, which is evident from the dielectric function. For this we have swapped Fig. 3(b) and SI-Fig. S5. The dielectric function separated into real and imaginary part is now presented in the main paper. The real permittivity becomes negative in a narrow spectral range in the vicinity of the excitonic transition due to its large oscillator strength. This “metal-like” feature is a topic on its own, greatly underexplored for organic matter, even though this phenomenon made plasmonics prominent (albeit of different physical origin). We added new references addressing this feature: Gentile et al. Nano Lett. 14 (2014) 2339. Gu et al. Appl. Phys. Lett. 103 (2013) 021104. Woo et al. ACS Photonics 4 (2017) 1138. See also response to Reviewer#2. However, to the best of our knowledge there is no literature available for an organic material combining “metal-like” negative real permittivity with excitonic circular dichroism. This combination makes up the originality of our manuscript. We strengthen this hypothesis by providing our own benchmarking with a *weakly coupling* excitonic material not showing a giant CD signal, see first comment to Reviewer#1.

As a further definitely minor comment to the rebuttal. I am not convinced of the statement: “the overall Fresnel reflection losses suffered by the transmitted beam are the same regardless of which side of the sample is illuminated first”, because to my (weak) understanding it depends on the difference in refractive indices: on the two faces of the sample, light enters the glass of the support from air or from the film and the two situations look very different to me. By far I am not expert in this field and for this reason, I more seek enlightening than criticize the authors’ work.

We are happy to provide further information. We remark that we refer to reflection losses, not to the overall reflectance which, as indicated by the referee, can be different due the differences in refractive indices. Accordingly, the absorption [note: not absorbance which is $-\log(\text{transmission})$] changes. We kindly invite the referee to check the paper E. D Palik et al., *Transmittance and reflectance of a thin absorbing film on a thick substrate*, Appl. Opt. 17 (1978) 3345-3347. To quote the first paragraph, “The reflectance and absorptance of a sample consisting of a thin metallic film on a thick glass substrate depend on the direction of incidence, i.e., whether the light enters from the film or the substrate side.¹⁻⁶ The transmittance is the same for both directions of the beam, however.” In particular, note that Eqs. (11) and (16) of this paper are equal, indicating that the transmittance (= transmission) and absorbance [$-\log(\text{transmission})$] for forward (11) and backward (16) propagation are the same:

$$\text{Eq(11): } T_u = \eta \tau_{15} T_{S1} / D$$

$$\text{Eq(16): } T_d = \eta \tau_{S1} T_{15} / D$$

Here, τ_{15} and τ_{S1} are the transmission coefficients through the interface between film and substrate for forward and backward propagation, respectively, and are identical to each other. The parameter $\eta = \exp(-\alpha L)$ depends on the substrate thickness L and its absorption coefficient α . T_{15} and T_{S1} consider the transmittivity of the film – substrate interface for forward and backward propagation, and D involves reflection coefficients, reflectivities, as well as thickness and absorption coefficient of the substrate.

Therefore, this indicates the losses regarding the transmission are the same. Our proposed reflection-correction is based on transmission/absorbance data which clearly is the same from both sides under normal incidence.

Reviewer #3

The paper has been considerably improved since the first submission, and the authors have responded well to the comments made. It remains that the main novelty of the study lies in the use of Mueller Matrix spectroscopy for determination of true Circular Dichroism. It should be noted that there are other studies using this approach that are not cited/referred to by the authors (e.g. Arturo Mendoza-Galván et al 2018 J. Opt. 20 024001).

Thank you for drawing our attention to the paper by Galvan and coworkers. It nicely shows that Mueller matrix spectroscopy can easily disentangle circular dichroism and Bragg reflection. We have gladly added the paper to our list of references. Note that this paper was published after we submitted the first version of our paper.

Indeed, Mueller matrix spectroscopy is rarely used for characterizing circular dichroism of organic semiconductor thin films. But for us it is a means to an end to unveil the extraordinary material's properties that otherwise would have been overlooked.

Technical comments:

Figure 6c is overly chaotic and should be improved.

The data for non-thickness normalized CD (Fig 6 (a) and 6(b)) should be provided in the Supporting Information.

I do not understand why the dissymmetry factor varies with layer thickness (the authors state this is obvious). Replicate values and associated error bars should be added to check this trend is indeed 'within experimental error', as stated.

We have revised Figure 6, see below. The as-measured CD data are removed from Fig.6 (c) and error bars have been added to the dissymmetry factor, Fig. 6(d). Only the *apparent* dissymmetry factor increases with increasing layer thickness as one can see in plot (d) (open squares and circles). But these are the uncorrected data, the thickness dependence is wrong. After the reflection-correction, the *true* dissymmetry factor is independent of layer thickness (filled squares and circles in (d)) as it is supposed to be.

Figure 6

We hope that you can agree on these arguments and will come to the conclusion that our manuscript can be accepted for publication in Nature Communications. Thank you very much for all your efforts in advance.

REVIEWERS' COMMENTS:

Reviewer #1 (Remarks to the Author):

I have read carefully the replies to referees, and am now satisfied that this work is ready for publication.

Reviewer #2 (Remarks to the Author):

The main concern I raised on the previous revision was convincingly solved. I also thank the authors for the comprehensive reply to the question I raised on light losses. Concerning the remarks of my colleagues, I am not sure to understand how did the authors chose NPTTPN as a "reference material". At the same time, however, I did not understand the request: "The present work would be aided enormously by showing just one other type of material under the same conditions". Finally, following my previous appreciation of this work, I would recommend publication.

Reviewer #3 (Remarks to the Author):

The authors have done everything that has been asked of them and the manuscript is much improved. With the addition of data for a benchmark material, I think this study is now acceptable.